

# 1 A method to predict the uncompleted climate transition
# 2 process

**3 Pengcheng Yan[1], Guolin Feng[2], Wei Hou[2]**

[1]{Institute of Arid Meteorology, China Meteorological Administration, Key
Laboratory of Arid Climatic Change and Reducing Disaster of Gansu Province, Key
Laboratory of Arid Climatic Change and Reducing Disaster of China Meteorological
Administration, China}
[2]{National Climate Center, China Meteorological Administration, China}
[*]Correspondence to: Wei Hou (houwei@cma.gov.cn)

## 10 Abstract

Climate change could be expressed as a climate system transiting from the initial state
to a new state in a short time. By considering the short period as a continued process,
which is called transition process, more details of climate change would be described
according to analysis the time sequence self. We had proposed a method to quantify
the transition process of the Pacific Decadal Oscillation(PDO) time sequence and
global sea surface temperature system. And the quantitative relationships among the
parameters characterizing the abrupt changes is revealed during the transition process.
In this paper, we develop this method to predict the end moment(state) if the transition
process has not been completed. Application of prediction method to the PDO
sequences indicates that the PDO index increased from a stable stage before 2011 and
gradually evolved to a transition process, and it was likely to end in 2015,which is
consistent with observations.

## 23 Keywords

Prediction method; Transition process of abrupt change; System stability; Pacific
Decadal Oscillation

## 26 1. Introduction

A system transiting from one stable state to another in a short period is called



abrupt change(Charney and DeVore, 1979; Lorenz, 1963, 1979). The abrupt change
system has two or more states(Goldblatt et al, 2006; Alexander et al, 2012), the
system swings between these states that is also called attractors in physics. This
phenomena is verified in many fields including biology, ecology, climatology(Thom,
1972; Overpeck and Cole, 2006), brain science(Sherman et al, 1981), human behavior,
etc. The latest famous climate change event is global warning hiatus, which has been
studied deeply by many researches(Amaya et al, 2018; Kosaka and Xie, 2013). Seven
different kind of abrupt changes are mentioned in Thom's research. And over the last
several decades, many methods have been proposed to identify different kinds of
abrupt change(Li et al, 1996), like Moving T-Test, Cramer's(Wei, 1999),
Mann-Kendall (MK, (Goossens and Berger, 1986)), Fisher (Cabezas and Fath, 2002),
etc. It is noticed that most abrupt change detection methods suggests that the abrupt
change is only a turning point, and the significant difference of a sequence on the two
sides of the turning point is defined as the index to measure the abrupt change. This
detection has a drawback. Notably, when the abrupt change occurs at the end of
sequence, it is difficult to detect. The filtering process considers the continuity of the
time series and provides a very important technique for the detection and prediction of
abrupt change. For an abrupt climate change event, the determination of the start
moment and end moment for the abrupt change is helpful for better understanding
abrupt climate change, and it is also an important supplement for studies of abrupt
climate change.

22       Mudelsee (2000) studied the abrupt change of a time sequence and illustrated

that abrupt change has a duration, namely, the transitional process. Moreover, it can
be quantitatively described with the Ramp Function, which is more advanced than
traditional metrics of abrupt change. Yan et al. (2014, 2015) developed the detection
method with a Non-linear Function replacing the Ramp Function. The new method
can confine the beginning and ending points of abrupt change and quantitatively
describes the process of abrupt climate change, and three parameters are introduced.
Besides, a quantitative relationship among the parameters is revealed (Yan et al, 2015).
The relationship could be used to predict the end moment(state) if the system had left





the original state but not yet got reach to the new state.
In this paper, several ideal time sequences are tested to study the prediction
method. Then, the method is applied to study the abrupt change of the Pacific decadal
oscillation (PDO), which is an important signal (Mantua et al, 1997; Zhang et al, 1997)
that reveals climatic variability on the decadal timescale. The PDO has not only affect
inter-decadal change in the Pacific and surrounding areas but also has a modulation
effect on the inter-annual signal (Yang et al, 2004). Previous studies (Lu et al, 2013;
Mantua et al, 1997; Trenberth and Hurrell, 1994) have indicated that there have been
many oscillations in the PDO over the past 100 years, and in particular, the most
famous are the transformations in the 1940s and 1970s. During the 1940s, the PDO
transformed from a high state to a low state, and during the 1970s, it did the opposite.
All this changes and their processes had been studied in the previous researches (Yan
et al, 2015; Yan et al, 2016). However, the research on the prediction of abrupt change
process is rare and difficult. Thus, based on the theory of abrupt change processes and
the continuity characteristics of the time series, we develop a new method to perform
a non-linear extrapolation to predict the end moment (state) of the abrupt change of
the PDO, by quantitatively identifying the abrupt change process of the PDO.
**2. Methods**
**2.1 The detection method of transition process**
The real time sequence change abruptly as shown in figure 1a, and the system
jumps to a high state in point C. If the period around point C is expanded to a longer
period, or the period around point C is observed on a more short time scale, a
transition period is obtained in figure 1b, which is a part of the original time sequence.
In fact, many abrupt change should be considered to be a transition period much more
than a point. Mudelsee(2000) indicated that this kind of abrupt change and the
transition period can be expressed with an ramp function. As shown in figure 1c, the
time sequence is divided into three segments, including two equilibrium states and
one increasing state. Then, the following ramp function was used to fit the transition




period.

$$x_t = \begin{cases} x_1 & t \le t_1 \\ x_1 + (t - t_1)(x_2 - x_1)/(t_2 - t_1) & t_1 < t \le t_2 \\ x_2 & t > t_2 \end{cases} \tag{1}$$

$t$ represents time, and $x_t$ represents its state. Based on the linear fitting method, $x_t$
was obtained to describe the system's behavior. By referring this work, a novel
method(Yan et al, 2015) was proposed to detect the transition period by using the
logistic model(May, 1976). The logistic model was created to study the population of
Humans or insects. The population changed( increased or decreased) looks like the
expanded of abrupt change, which is also like the change in figure 1c. The modified
logistic model is expressed as follows and its image is shown in figure 1d:
$\dot{x} = k(x - u)(v - x) \tag{2}$
Parameters $u$ and $v$ represent the two equilibrium states respectively. Parameter $k$
represents the speed of switching between different states. As shown in figure 2a,
parameters $u$ and $v$ being fixed, and setting $k$ as 0.5, the system transiting to the new
state costs a shorter time than that setting $k$ as 0.4. If parameter $k$ is set large enough,
the system collapses and becomes chaotic( as shown in figure 2b). When parameter $k$
is set to different values, more situations have been discussed in detail in the previous
research(Yan et al, 2016). The result shows that parameter $k$ characterizes the stability
of the system (the larger the absolute value, the more unstable the system). Besides,
the system does not always evolve to one of the two stable states. If the initial state of
the system is between two stable states, the system will converge to one of states;
otherwise, the system will crush.
Mathematical derivation is an effective method to learn the characteristics of the
system of the logistic model. The left side of Eq.(2) represents the change of variation
with time, and it can be considered to be the general velocity, which is also the
derivative of variation to time. We continue to calculate the derivative of velocity to
time, and we have the general force as follows:
$\ddot{x} = 2k^2 [x - (u + v)/2](x - u)(x - v) \tag{3}$





Calculating the spatial integral of general force (the system itself), the general
potential energy is obtained as follows:

$$V_{(x)} = -\int_0^x \ddot{x}dx = -\int_0^x 2k^2[x-(u+v)/2](x-u)(x-v)dx$$
$$= \frac{k^2}{2}[x^4 - 2(u+v)x^3 + (u^2+v^2+4uv)x^2 - 2(u+v)uvx]$$

3                                                                                              (4)

According to Thom's(1972) theory, the system described by a quadratic function
would exhibit tipping-point abrupt change, which the system jumps from one state to
a new state abruptly. In figure 2c, the potential energy of Eq.(4) is verified to have two
states with the lowest energy, which are stable, and the system is able to transit
between them. This bistable structure is common in the climate system (Goldblatt et
al, 2006). Therefore, Eq.(2) can be used to describe the abrupt change system, and the
parameters represent different key factors of the transition period during abrupt
change.
In figure 2d, the smooth continuous time sequence is divided into three segments.
In each segment, the parameters can be obtained by regression. For the first segment,
the system is stable, and the average value of system states is $v$. Similarly, in the third
segment, the average value of system state is $u$. Parameters $v$ and $u$ are the two stable
states of the system, which their values can be calculated by Eq.(5), where $n_1$ and $n_3$
are the length of first segment and the third segment respectively. For the second
segment, the transition period can be considered linear approximately. Then, the linear
trend $h$ is calculated by regression (Huang, 1990) in Eq.(6), where $i$, $x_i$ denote the time
and the state of the system at this time, and $\bar{i}, \bar{x}_i$ are their averages respectively. And,
$n_2$ is the length of second segment.

$$\begin{cases} v = \sum_{i=1}^{n_1} x_i / n_1 \\ u = \sum_{i=n_1+n_2+1}^{n} x_i / n_3 \end{cases}$$
                                                                                              (5)

$$h = \sum_{i=n_1+1}^{n_1+n_2} \bar{i} \cdot \bar{x}_i \bigg/ \sum_{i=n_1+1}^{n_1+n_2} \bar{i}^2$$                    (6)
Additionally, the linear trend $h$ can be expressed with two points on the curve



approximately as follows, where the two points are represented by $A(x_a, t_a)$ and $B(x_b,$
$t_b)$(figure 2d).
$$h = \frac{x_a - x_b}{t_a - t_b} \qquad (7)$$

4       Then, we do integration for both sides of Eq.(2). In order to simplify Eq.(8), an

intermediate variable( Eq.(9)) is introduced, and Eq.(8) is rewritten as Eq.(10).

$$\frac{dx}{dt} = \kappa(x - \mu)(v - x)$$
$$\Rightarrow \frac{1}{\mu - v} \int_{x_0}^{x} \left( \frac{1}{x - v} - \frac{1}{x - \mu} \right) dx = \kappa \int_{t_0}^{t} dt$$
$$\Rightarrow \ln\left( \frac{x - v}{x - \mu} \right)\left( \frac{x_0 - \mu}{x_0 - v} \right) = \kappa(\mu - v)(t - t_0)$$
$$\Rightarrow \frac{x - v}{x - \mu} = \frac{x_0 - v}{x_0 - \mu} e^{\kappa(\mu - v)(t - t_0)}$$
$$\qquad (8)$$

$$\xi(t) = \frac{x_0 - v}{x_0 - \mu} e^{\kappa(\mu - v)(t - t_0)} = \frac{x - v}{x - \mu} \qquad (9)$$
$$t = t_0 + \frac{1}{(\mu - v)\kappa} \ln \frac{x_0 - \mu}{x_0 - v} \xi(t) \qquad (10)$$

9       The transition period including points $A$ and $B$ is approximately linear. Thus, the

following relationship is established by defining the location parameters $\alpha, \beta$.
$$\begin{cases} \alpha = \frac{x_a - v}{\mu - v} \\ \beta = \frac{x_b - v}{\mu - v} \end{cases} \Rightarrow \begin{cases} x_a = \alpha(\mu - v) + v \\ x_b = \beta(\mu - v) + v \end{cases} \qquad (11)$$
Then, Eq.(10) and Eq.(11) are introduced into Eq.(7), and the difference between
the two stable states( $u$ and $v$) is defined as the amplitude of change($w$). A relationship
among the linear trend($h$), stability parameter($k$), amplitude of change($w$), and the
location parameters($\alpha, \beta$) are confirmed.





$$h = \frac{[\beta(\mu-\nu)+\nu]-[\alpha(\mu-\nu)+\nu]}{\frac{1}{(\mu-\nu)\kappa}\left(\ln\frac{x_0-\mu}{x_0-\nu}\xi(t_\beta)-\ln\frac{x_0-\mu}{x_0-\nu}\xi(t_\alpha)\right)}$$

$$= \frac{\kappa(\mu-\nu)^2(\beta-\alpha)}{\ln\frac{\xi(t_\beta)}{\xi(t_\alpha)}} = \kappa w^2 \frac{(\beta-\alpha)}{\ln\frac{\beta(\alpha-1)}{\alpha(\beta-1)}}$$

(12)

The part$_,$ $(\beta-\alpha)/\ln((\beta(\alpha-1))/(\alpha(\beta-1)))$ $_,$ in Eq.(12) is only related to the
location parameters, then let it be $\chi$, and the relationship among $\chi, \alpha, \beta$ is displayed in
figure 3a. Then, the relationship of Eq.(12) is rewritten as Eq.(13):
$h = \kappa\omega^2\chi$ (13)
According to the numerical experiment, the relationship between parameter $\chi$
and location parameters is shown in figure 3a and figure 3b, and figure 3a is the
profile of the diagonal in figure 3b, which indicates that the sum of $\alpha$ and $\beta$ is 1.
Parameter $\chi$ changes little when the location parameter varies in a certain range as
shown in figure 3b. It is obvious that the closer the points( A and B) are to the turning
points, the more the process between point A and point B can represent the whole
transition process as shown in figure 3c. However, the process between point A and
point B is linear when the two points are located to the top of the segment. Let the
sum of $\alpha$ and $\beta$ be 1, then the change of parameter $\chi$ is only related to parameter $\alpha$(or
parameter $\beta$), as shown in the diagonals in figure 3b( also in figure 3a). Parameter $\chi$
changes little when parameter $\alpha$ is about 0.2 or larger. In figure 3c, three ideal
experiments were carried out. In each experiment, points($A$ and $B$) were set to be
different positions, and their parameters were calculated respectively as table 1. The
parameters $\alpha$ are set as 0.20/0.25/0.15 respectively in three different tests. For test 2
and test 3, both of the percentages of $\alpha$ changing to test 1 are 25%, while the
percentages of $\chi$ changing are only 4.07% and 7.73% respectively, which means the
percentage change of $\chi$ is much less than $\alpha$. In addition, linear trends of these three
ideal models are calculated according to the points and by regression method, and the
values are shown in table 1. It is noted that although the positions of points are
different, the trend obtained according to the points is almost the same as that




obtained by regression method. The error percentages are 2.36%, 2.25%, 1.38% respectively, which means that when the position of the points(the values of parameters $\alpha$ and $\beta$) changes slightly, there is little influence on the parameters.

**2.2 The prediction method of transition process**

Eq.(13) shows the quantitative relationship among linear trend, stability parameter, and amplitude of change. There is a linear relationship between linear trend and stability parameter; and there is the quartic function relationship between linear trend and amplitude of change. We did reveal this quantitative relationship much more than in theory but in real time series by studying the sea surface temperature (Yan et al, 2016). Based on this relationship, we are going to improve a method to deal with the problem which the transition process is not finished. During the real time sequence, the system transits away from the original state, but it has not been in a new state as shown in figure 4. The red line represents the period which has been experienced, while the gray line represents the period which hasn't been experienced. Based on the system states which is far away from the original state, a quasi linear extension of the transition process is established(dash line). Then the parameters $v$ and $h$ are obtained by the method of Eq.(5) and Eq.(6). Assuming that the parameter $k$ satisfies the statistics in the history of the system, the parameter $u$ can be predicted on the basis of Eq.(13), and the end moment of this transition process is also predicted apparently. An ideal time sequence is constructed by using the logistic model and random numbers to achieve the prediction as shown in figure 5a, and three uncompleted changes are shown in figures 5b, 5c and 5d respectively. The parameters $v$, $u$ and $k$ of the logistic model are set as -1.0, 2.0, 0.1, for the ideal time sequence, and the random number range is 0-1. The parameters $v$, $h$ are obtained by regression method when making prediction. There is no way to obtain the parameter $k$ from the historical data for an ideal sequence, thus it is given directly, and the prediction of the end state( moment) is drawn in the graph( represented by blue line). The results show that this prediction method performs well even the system stays in different position of the transition process.





## 3. Prediction studies of the parameter threshold and the index for the PDO abrupt change process

In order to test the validity of this prediction method in a real climate system, the Pacific Decadal Oscillation(PDO, which represent the interdecadal variability of the Pacific, (Newman et al, 2016)) index is used to detect the transition process over the past 100 years. The PDO index data used is from website of the University of Washington (http://research.jisao.washington.edu/pdo/). It's found that PDO index gradually increased in 2011. Therefore, it is necessary to study the end moment(state) of this transition process. The time period from January of 1900 to November of 2015 is studied as the training data, and the time period from December of 2015 to April of 2017 is used as the test data.

### 3.1 Threshold of stability parameter $k$

The histogram in Figure 6a shows the PDO time sequence from January of 1900 to November of 2015, and it shows that the PDO went through several changes. The green dots in Figure 6a are parameter $k$ when the sub-sequence length takes 20 years. In the early 1940s and late 1970s, there are two abrupt changes of the PDO, and the absolute value of the stability parameter $k$ is large, which means that the system is not stable during transition periods. In the 1940s, the PDO transits from a positive phase to a negative phase, and the $k < 0$, whereas the situation in the 1970s is the opposite. Figure 6b shows more $k$ values corresponding to the different sub-sequence lengths (as indicated by X-axis, the variation range of the sub-sequence is 20-60 years, with an interval of 1 year). The Y-axis is the start moment of abrupt change, and the locations of the dots indicate the start moments of detected abrupt changes for the corresponding sub-sequence lengths. In particular, the blue dots represent that parameter $k$ is negative, and the red dots represent that it is positive. The dots in the left side region are clearly denser than in the right side region. This is because when the length of sub-sequence is short, the magnitude of abrupt change is also often small.



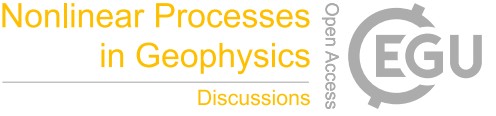

Therefore, for the entire sub-sequence, there are many detected abrupt changes. When
the length of the sub-sequence reaches or exceeds 50 years, the detected abrupt
change mainly begins in the 1940s and 1970s. This is also the abrupt change that has
been investigated in other research (Shi et al, 2014). The abrupt changes in these two
periods correspond to large $k$ values, which means that these two abrupt changes are
more unstable than others. More statistical results indicate that the threshold
distribution of parameter $k$ values in historical abrupt change processes exhibit
multiple peaks (Figure 7). Specifically, the peak with the largest probability is located
near to 0. The $k$ value in the distribution is small, which indicates that the abrupt
changes that correspond to these $k$ values are stable. The $k$ values are distributed in the
peaks on the left side and right side of the origin. When $k<0$, the PDO time sequence
transits from the positive phase to the negative phase, when the threshold of the $k$
peak is wide and the probability is small; when $k>0$, the PDO time sequence
transforms from the negative phase to the positive phase, when the threshold of the $k$
value is narrow and the probability is large. This indicates that the two transitions,
which one of them is that the system changes from the positive phase to the negative
phase, and the other is that the system changes from the negative phase to the positive
phase, are not symmetric, and the latter is more stable. Because there is a difference in
parameter $k$ when the selected sub-sequence length is different, the gray region in the
upper right corner of Figure 7 also shows the statistical properties of parameter $k$
when the sub-sequence length is 20, 30, 40, 50, or 60 years. When the length of the
sub-sequence is 20 years and 30 years, there is only one peak in the distribution of $k$
values, and the parameter $k$ value of the peak is about 0, which means that the abrupt
change is stable. That is, when the time sequence is short, the detected amplitude of
the abrupt change is small and stable. It is difficult to detect an abrupt change with
huge amplitude if the time scale is tiny. When the length of the sub-sequence is 40, 50,
or 60 years, the peak value on the side of $k>0$ is not considerably different, which
indicates that the stability degree of the abrupt change from negative to positive is
consistent; the location of the peak value on the side of $k<0$ moves to the left as the
sub-sequence length increases, which means that the sub-sequence is longer, the





amplitude of detected abrupt change is larger, and it is more unstable. From the
perspective of the value, a $k$ value in the range of (-10, 10) accounts for 80.2%, a $k$
value in the range of (-5, 5) accounts for 74.2%, and a $k$ value in the range of (-2, 2)
accounts for 58.6%. In the following studies, the $k$ value is mainly set in the range of

5    (-2, 2).

**3.2 Determination of abrupt change and the threshold for the initial**
**state $v$ and linear trend $h$**
We use the method to analyze the abrupt changes occurred in the past 10 years.
In particular, the abrupt changes that began in 2007 and 2011 have been identified.
Parameters $v$ and $h$ are obtained with sub-sequences of 10, 20, 30, or 40 years for two
abrupt changes (Table 1). Specifically, on the scales of 10 and 20 years, we first
detected an abrupt change beginning in 2011, and the states were -0.45 and -0.03,
respectively, with an linear trend of 1.054/a. On the scales of 30 and 40 years, the
earliest detected abrupt change began in 2007, and the states before the abrupt change
were 0.36 and 0.41, respectively, with an linear trend of 0.227/a. The abrupt change
detected on the scale of 50 and 60 years began in 1976, and therefore, it is not
investigated. The results above validate that if we select sub-sequences of different
lengths, the detected abrupt changes are also different. This is because if the length of
the sub-sequence is different, the probability distribution of the magnitude of detected
abrupt change process in the sequence is different. Therefore, the abrupt change
determined through the percentile threshold method is also different, which indicates
that the abrupt change has some scaling properties.
Figure 8 shows the segment of PDO around the abrupt change points, which
contains the different stages of the abrupt change process. The time sequence has a
period of steady state before the abrupt change begins and then enters the
development state and gradually transforms to the new steady state. We first analyse
the abrupt change that began in 2007. As shown in Figures 8c and 8d, before and after
2007, the time sequence transforms from the stable to an augmented abrupt change. It



is worth noting that due to the different sub-sequence selections, the abrupt change began during the previous stable state in 2007, and Figures 8a and 8b show different results. However, after the abrupt change begins, the rate of abrupt change during the persistent process of abrupt change is consistent, which further supports the detection result that is shown in Figure 9. For the abrupt changes beginning before and after 2011, before the abrupt change begins, the sequence maintains stability over some time and then begins to gradually increase (Figures 8a and b). The abrupt climate changes of two PDO sequences beginning in 2007 and 2011 both belong to the augmented abrupt change.

In the three abrupt changes that were obtained by the aforementioned analysis, the sub-sequence length interval is 10 years, and we further select the interval of the sub-sequence to be 1 year. That is, we respectively study the abrupt change detected under the situation when the sub-sequence length is selected as 10, 11, 12, ⋯, and 40 years, and we study the initial stage $v$ of these abrupt changes and the linear trend $h$. Figure 9 shows that although the interval of the sub-sequence length is shorter, the detected abrupt change only has two abrupt changes. One began in 2007, and the other began in 2011. Moreover, every linear trend $h$ is the same (the specific values are shown in Table 1), whereas the $v$ values of the initial stage are different. In particular, the abrupt change that began in 2007 is detected for the sub-sequence of 30-40 years, and the value of parameter $v$ is in the range of (0.28, 0.45). The abrupt change that began in 2011 is detected for the sub-sequence of 10-30 years, and the value of parameter $v$ increases as the length of the sub-sequence increases, whereas the variation range of threshold is (-0.48, 0.12), which is significantly different from the situation in 2007.

## 3.3 Prediction study on the turning of the abrupt change beginning in 2011

After the threshold ranges for parameters $k, v,$ and $h$ are determined, according to the quantitative relationship, we can calculate the stable state and the end moment of




the abrupt change after the abrupt change of the system, and we therefore predict the

turning of a sequence. Using the abrupt change in 2011 as an example, we study the

ending state and end moment for the PDO index abrupt change. According to the

research results that are presented in Sections 3.1 and 3.2, the parameter is $h$=1.054/a

in this abrupt change, and the threshold range of parameter $k$ is determined to be (0, 2).

The range of parameter $v$ is determined to be (-0.48, 0.12), and the variation situation

of parameter $\mu$ and end moment with parameters $k$ and $v$ are shown in Figure 10. The

results indicate that the threshold range of parameter $u$ for the ending state is (1, 7),

and the time range of the ending moment is (2013a, 2017a). Because of the

probability distribution function of stability parameter $k$, the probability for this abrupt

change to end after 2015 is large, and after 2015, the sequence stops the augmented

transformation, approaching stability.

Figure 11 shows the PDO time sequence(black lines) and the prediction

result(black dot dash lines). The trend of the PDO time sequence during 2006-2011 is

almost 0, which means that the period belongs to the stable stage prior before the

abrupt change point. After 2011, the sequence increases significantly, and it is not able

to be known whether the increasing process has been completed or not. The prediction

result show that the abrupt change process has been completed in 2015. And during

2016-2017, the trend of the index is almost 0 too, which means that the system stops

to increase and it reaches to a new stable state. The real PDO sequence(star points) is

consistent with the prediction results.

## 4. Conclusion and discussion

In this paper, we develop a method to study the uncompleted transition process.

The method is applied to predict the ideal time sequences and the PDO time sequence.

The quantitative relationship between the parameters characterizing the transition

process plays an important role. For the PDO time sequence, the abrupt change started

began in 2011, and the end moment is predicted to be 2015, which is consistent with

the real time series. The study indicates that the PDO system over the past 100 years





exhibits a bistable structure, which is also be mentioned in the previous findings(Yan et al, 2016). It means that the sea surface temperature system in the Pacific often changes between different states. Besides, the detection of abrupt change being consistent with timescale is revealed, namely, there are some differences in the abrupt changes that are detected on different timescales in which the most recent abrupt change on the 10-20-year scale began in 2011. In traditional researches of time sequence abrupt change, time scale is rarely mentioned, while the same abrupt change is detected as different forms. The abrupt change with smaller time scales has a continuous process, and the abrupt change with larger time scales becomes abrupt change point.

In this paper, a detected abrupt change beginning in 2011 appears relatively close to the end of the 115-year sequence, and it is difficult to identify by using other methods. However, the method we proposed not only detects abrupt changes that occur at the end of the sequence but can also do prediction. The findings increases the possibility of resolving the problem associated with difficult processing at the end of a time sequence, and it also provides new thoughts and a new method for studies regarding the prediction of abrupt changes.

## Acknowledgements

This study was jointly sponsored by National Key Research and Development Program of China (Grant No. 2018YFE0109600), National Natural Science Foundation of China (Grant Nos. 41875096, 41775078, 41675092), Northwest Regional Numerical Forecasting Innovation Team (GSQXCXTD-2017-02).

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





Table1. The parameters of ideal models

|  | $\alpha$ | $\chi$ | h0 | h | \|h0-h\|/h |
|---|---|---|---|---|---|
| test 1 | 0.20 | 21.87E-2 | 12.99E-4 | 12.69E-4 | 2.36% |
| test 2 | 0.25 | 22.76E-2 | 9.10E-4 | 8.90E-4 | 2.25% |
| test 3 | 0.15 | 20.18E-2 | 32.27E-4 | 32.72E-4 | 1.38% |

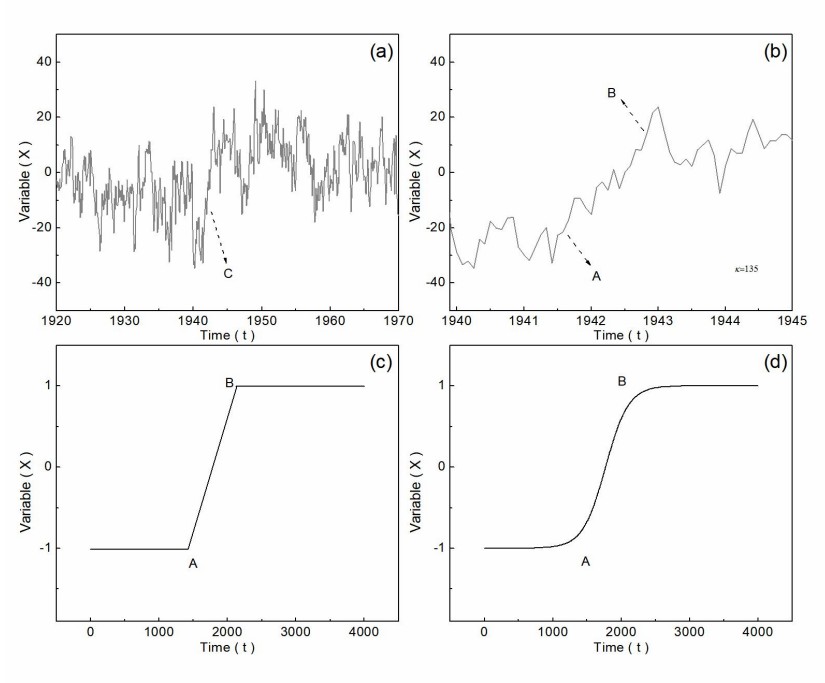

Figure 1. Transition process of abrupt change in real time sequence and ideal time
sequence. (a) The PDO time sequence during 1920 to 1970; (b) The PDO time
sequence during 1940 to 1945; (c) The transition process presented by linear function;
(d) The transition process presented by nonlinear function



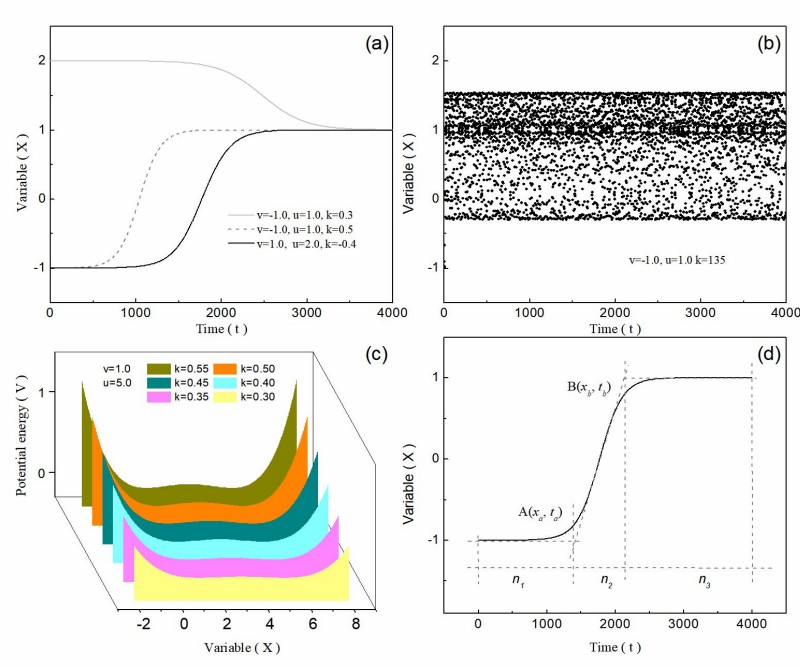

Figure 2. The system presented by Eq.(2). (a)The system stays in stable states since
the parameters are different; (b)The system stays in unstable states since the
parameters are set as some values; (c)The generalized potential energy function of
system performs differently since the parameters are different; (d)Different segments
of the transition process in the ideal time sequence

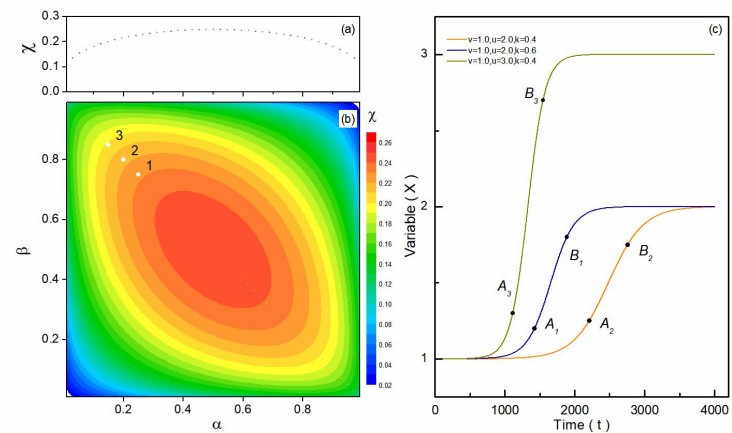

Figure 3. The influence of different value of parameters $\alpha$ and $\beta$ on parameter $\chi$ and
parameter $h$. (a) Diagonal section of parameter $\chi$ in figure b; (b) Parameter $\chi$ with
parameters $\alpha$ and $\beta$; (c)Points $A$ and $B$ stay in different positions in three tests.

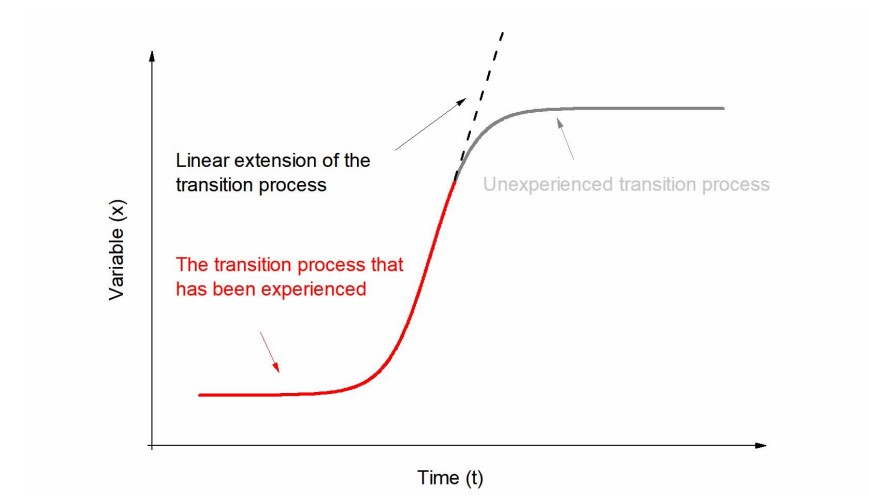

Figure 4. The schematic diagram of prediction method

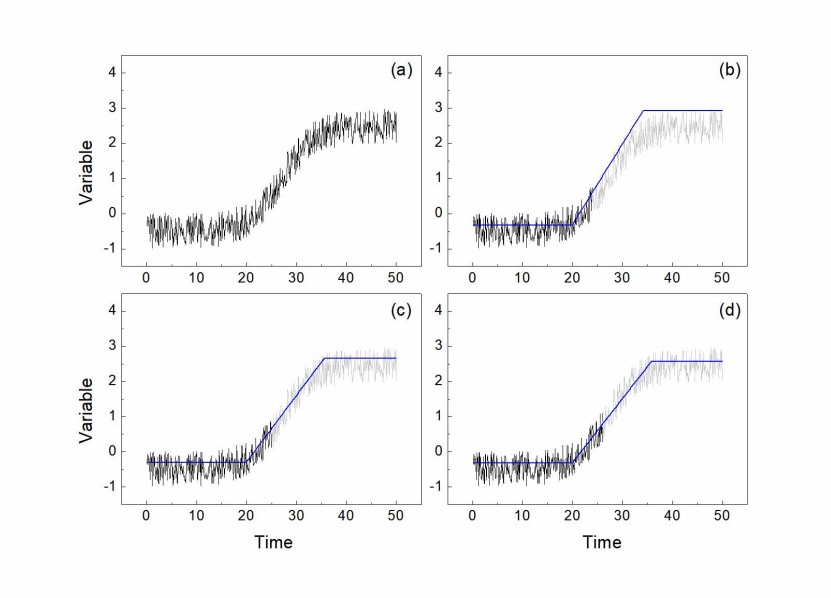

Figure5. The ideal time sequence constructed by the logistic model and random
numbers. (a) Completed transition process with length of 50, Uncompleted transition
processes(the gray lines) and their prediction result(the blue lines) with length of
25(b), 26(c), and 27(d)


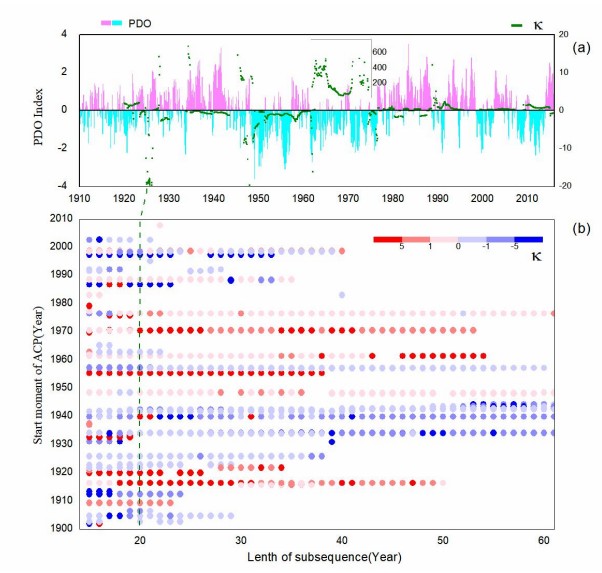

Figure 6. Identification of the PDO time sequence and stability parameter k during an
abrupt change under different sub-sequence lengths. (a) The X-axis is the year, the
histogram in the figure shows the PDO time sequence (left panel), and the green dots
indicate the magnitude of parameter k when the sub-sequence is 20 years (right panel);
(b) the identification of abrupt change and the stability parameter identified for
different sub-sequence lengths (pink indicates augmented abrupt change, and blue
indicates decreasing abrupt change, with deeper colors representing higher values).
The X-axis is the sub-sequence length (month), and the Y-axis is the beginning time
of abrupt change (year).

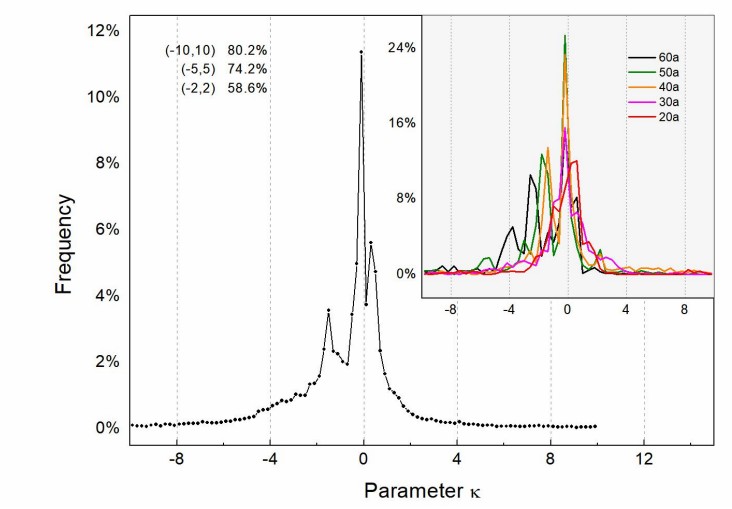

Figure 7. Statistical results of stability parameters for different sub-sequence lengths.
The X-axis is the value of the parameter, and the Y-axis is the statistical frequency
with a sub-sequence length of 10 years. The gray region in the upper-right corner is
for a sub-sequence of 20-60 years.

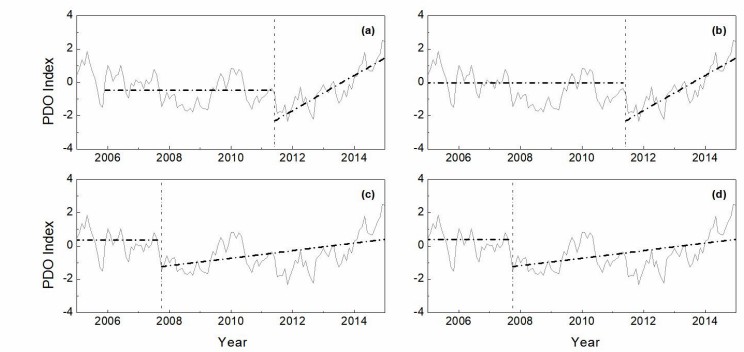

Figure 8. The PDO time sequences and the detection of parameters $v$ and $h$ when the
sub-sequence was set at (a)10 years, (b)20 years, (c)30 years, (d)40 years. The gray
lines is PDO time sequences. The horizontal dash lines are stable states before climate
change point, the slope dash lines represent transition processes of climate change,
and vertical dotted line is the start moment of climate change.

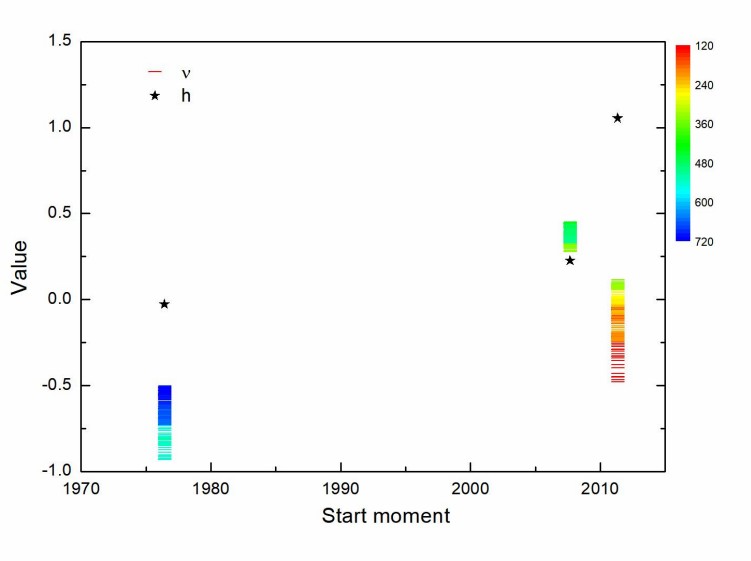

Figure 9. Several abrupt changes detected when the sub-sequence was set at different
lengths, as well as the values of the initial state $v$ and linear trend $h$ for the abrupt
changes. The black asterisk represents parameter $h$, and the colourful short bar
represents parameter *v*. The colour bar represents the sub-sequence length.

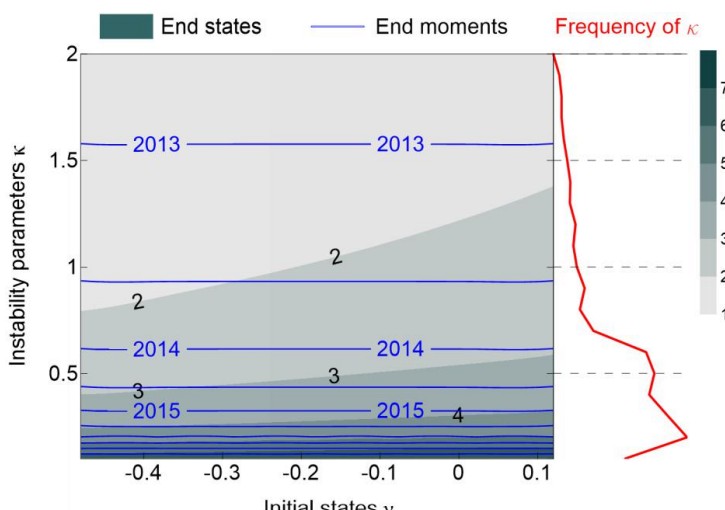

Figure 10. Variation ending state and ending time for a system with the initial state
parameter *v* (horizontal ordinate) and stability parameter *k* (vertical coordinate). The
red line on the right side shows the probability distribution of stability parameter *k*.

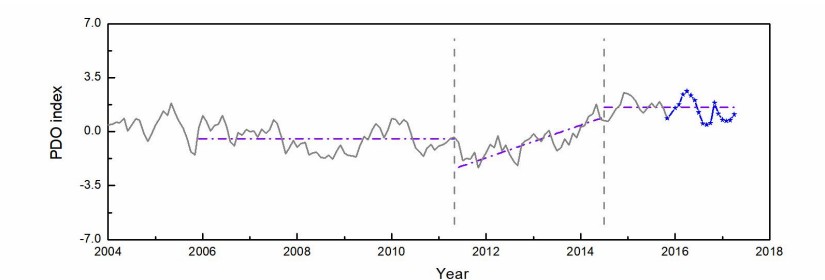

Figure 11. Prediction of the PDO index turning point. The gray solid line is the actual
PDO index used in the study until November of 2015; the solid line with an asterisk is
the actual PDO index from December 2015 to April 2017; the gray vertical dashed
line is a diagram for the beginning and ending times of abrupt change, which are 2011
and 2015, respectively; the dot-dashed line represents the system state at different
stages.