# Peer review of "A method to predict the uncompleted climate transition"

_Nonlinear Processes in Geophysics, 2020_

## Referee Comment (RC1) · Anonymous Referee #1 · 28 Feb 2020

General comments

The paper needs major changes to address all issues. The developments in Subsection 2.1 from page 4 to page 7 line 5 already appear in the first part of section 2 of ref. "Yan PC, Feng GL, Hou W. A novel method for analyzing the process of abrupt climate change. Nonlinear Processes in Geophysics 2015; 22:249-258, doi: 10.5194/npg-22-249-2015" pages 250-251 and do not introduce any new information. They should be omitted and cited or resumed.

The method to determine the values of location parameters $\alpha$ and $\beta$, or the position of points A and B is not clearly specified.

The numerical tests of section 2.1 are not fully specified neither its purpose.

The results of simulated prediction method of section 2.2 are drawn in figure 5 but not quantified in the text, so the quality of the prediction method can not be appreciated.

The method described in the paper is based on the use of continuous functions: piecewise linear functions or logistic model; but it is applied to discontinuous functions: see figures 8 and 11. The lines have jumps and the application of the previous equations to discontinuous functions must be justified.

The table with the results of analysis of past 10 years in Section 3.2 is missing.

Specific comments

2.1 The detection method of transition process Page 3, lines 21-22. "... the period around point C is expanded to a longer period, or the period around point C is observed on a more short time scale ..." It is not the same. Figure 1b corresponds to the second option: "observe on a more short time scale" or better "observe on shorter time scale". The idea is that with a more detailed view, the transition process can be observed.

In page 6 line 13 and eq. (12), the amplitude of change is denoted by w, but in eq. (13) the notation is changed to $\omega$.

Page 7, line 6 "According to the numerical experiment...". Figure 3 is not a numerical experiment; part (b) is a contour map of $\chi$ for $0 \leq \alpha \leq 1$ and $0 \leq \beta \leq 1$ and part (a) is a section of that contour map along the line $\alpha + \beta = 1$ (probably).

Page 7, line 8. The assertion "the sum of $\alpha$ and $\beta$ is 1", does it mean that figure 3(a) is the profile of figure 3(b) along the diagonal $\alpha + \beta = 1$? Please, clarify. See the remark for the caption of Fig. 3.

Page 7, lines 13-15. "Let the sum of $\alpha$ and $\beta$ be 1, then then the change of parameter $\chi$ is only related to parameter $\alpha$ ... (also in figure 3a)". This is obvious, $\chi$ depends on the two parameters (has two degrees of freedom), by imposing a relationship between the two parameters, you reduce the degrees of freedom to one. This sentence does not add any information and should be suppressed.

Page 7, line 16. "In figure 3c, three ideal experiments were carried out...". The experiments were not carried out in the figure 3c. Figure 3c describes the parameters of the three experiments. As noted before, the experiments deserve their own subsection.

The setup of the tests is not clearly described. From the figure 3c, we know the parameters u, v and k, from figure 3c. Other parameters are in Table 1. The test setup will be clearer if Table 1 would include all parameters for each test. Parameter h0 in Table 1 is not defined. Nothing is said about the time span of tests; if the points are randomly perturbed and how. See the previous remark.

2.2 The prediction method of transition process

Page 8 line 7, "... there is the quartic function relationship between linear trend and amplitude of change." Eq. (13) reads h=k $\omega$2 $\chi$. This equation is quadratic in the amplitude $\omega$, not quartic. Page 8, line 18, We are supposing the repetition of events, assuming all events have the same k. We obtain $\nu$ and h. Which is the value of $\alpha$? We are also assuming that $\alpha+\beta$=1, so we can calculate $\chi$.

Page 8, line 23, An ideal time sequence is constructed with a logistic model with parameters v=-1, u=2 and k=0.1. But in figure 5, the steady part of the curve is well above -1 in the left part and above 2 in the right part. From that graphics, the limits seem to be v=-1.5, u=2.5.

3.1 Threshold of stability parameter k

Page 9, line 15. The origin of the values for the parameter k (green dots) that appear in figure 6a is missing.

3.2 Determination of abrupt change and the threshold for initial state $\nu$ and linear trend h

Page 11, line 8, "We use the method to analyze..." But the method is not specified.

Page 11, line 10, "Parameters $\nu$ and h are obtained ... abrupt changes (Table 1)." But

the title of Table 1 is "The parameters for ideal models" and also written in page 7, line 18. Is there another missing Table?

Page 11, line 21. "...the abrupt change determined through the percentile threshold method ...". This method must be described or referenced.

Page 11, line 23. Along the paper, time series were approximated by piece-wise continuous functions: the system was in a steady state, and from that value changed up or down. But in figure 8 time series are approximated by functions with jumps from the value of the steady states to the beginning of the slope lines that approximate the changes. These profiles are different from those used in figure 5 to simulate the process of recovering the parameters and those of the logistic functions.

Page 13, line 4, "...the parameter h=1.054/a..." The units of h are not clear, what does a mean? Year? The same problem appears in line 9 in the same page.

Figure 3 Caption. (a) part: it is not stated which diagonal of (b) refers to, $\alpha = \beta$ or $\alpha + \beta = 1$. It, also, would be interesting to mark points 1, 2 and 3 from (b) in the part (a) of figure.

Technical corrections

Cites should be separated from text by a blank, e.g. p. 2 line 1 "... change (Charney ...)" and many more. Page 6 line 4 "Then, we do integration...", I consider better "Then, we integrate..."

---

## Referee Comment (RC2) · Anonymous Referee #2 · 23 Mar 2020

This article needs major revisions before it can be considered for publication. There are many issues with consistency, clarity, grammar, and the discussion of the major results. I will outline my concerns below.

**General comments**

1. The abstract needs to be made more clear. The phrase "more details of climate change" is too broad and does not explain what exactly is being addressed by the methods presented in the paper. The PDO is also not explained, nor its relation to climate change.

2. There is not enough introduction to the methods section before discussing the details of time series analysis.

3. The mathematical notation is inconsistent and unclear.

   - Variable $k$ appears to be often interchanged with $\kappa$
   - Parameter $k$ is referred to as both a stability (Fig. 10 and Section 3.3) and instability parameter (Fig. 10).
   - $\mu$ is never formally introduced and is potentially being exchanged with $u$.

4. There is terminology that is used but not defined.

   - continued process (pg 1, line 12)
   - the filtering process (pg 2, line 16)
   - ramp function (pg 2, line 24)
   - crush (pg 4, line 21)
   - percentile threshold (pg 11, line 21)
   - augmented abrupt change (pg 11, line 28)

5. There is inconsistency between Section 2.2 and Section 3. In Section 2.2 it is stated that the parameter $k$ cannot be obtained from the data (with no explanation as to why), so it is fixed *a priori*. In Section 3 the parameter $k$ has been estimated from a time series, but again with no explanation as to how one would estimate this.

6. In Section 3.1 it is stated "When the length of the sub-sequence is 20 years and 30 years, there is only one peak in the distribution of $k$ values..." (pg 10, lines 21-24). This seems strange, as there are said to be multiple peaks for a smaller subsequence (10 years), a single peak for 20 and 30, and then multiple peaks for larger subsequences. I would assume there would be a more continuous relationship. This is not discussed why this is not the case. Also, a quantitative measure is not specified of what defines a peak.

7. The motivation for Section 3.2 is absent and it is not obvious how this section relates to the overall goal of Section 3.

8. "Abrupt change" appears to be used synonymously with "transition process" in Section 3.2 and this does not seem consistent with the rest of the paper. Please maintain the same terminology for clarity.

9. The final paragraph of Section 3.2 (pg 12, lines 10-24) discusses three abrupt changes. The previous paragraph discussed four. There is much confusion as to what abrupt change events are being discussed throughout this paragraph.

10. The lengths of the subsequences mentioned in Section 3.2 do not match the numbers on the colour bar in Fig 9. It is therefore not clear what Fig 9 is showing.

11. There is no discussion as to which abrupt change detection (year 2007 or 2011) is correct, which leads to a lack of motivation for studying only the 2011 event. It needs to be more clearly explained why the 2011 event is chosen for the prediction experiment.

12. The "variation situation of parameter $\mu$" (pg 13, lines 6-7) was never introduced nor explained.

13. The "prediction result" (pg 13, lines 13-14) was not specified. Additionally, it is not clear which prediction is being shown in Fig. 11.

14. Conclusion needs to be expanded upon much more.

    - The sentence "The abrupt change with smaller time scales has a continuous process, and the abrupt change with larger time scales becomes abrupt change point." (pg 14, lines 8-10) is not easily understandable and alludes to material that does not appear to have been discussed in the paper.

- The phrase "a detected abrupt change beginning in 2011 appears relatively close to the end of the 115-year sequence, and it is difficult to identify by using other methods" (pg 14, line 11-13) was not previously discussed in the manuscript. Please expand on why the abrupt change is difficult to identify through other methods.

- There is not enough evidence in the manuscript to support the claim "The findings increases the possibility of resolving the problem associated with difficult processing at the end of a time sequence" (pg 14, lines 14-16). Please add discussion of the problem of difficult processing at the end of a time sequence.

**Specific comments**

1. pg 1, line 14 - Change "self" to "itself"

2. pg 1, line 15 - Add full reference to paper on PDO

3. pg 1, line 16 - Remove "And" from beginning of sentence

4. pg 2, lines 4-6 - Add references for each of the fields mentioned

5. pg 2, line 6 - Change "famous" to "observed"

6. pg 2, line 8 - Add reference for "Thom's research"

7. pg 2, line 8 - Remove "And" from beginning of sentence

8. pg 2, line 24 - "Ramp Function" does not need to be capitalised

9. pg 2, line 26 - "Non-linear Function" and "Ramp Function" do not need to be capitalised

10. pg 2, line 29 - Remove "Besides" from beginning of sentence

11. pg 3, line 1 - Change "got reach to" to "reached"

12. pg 3, line 3-4 - Capitalise "decadal oscillation"

13. pg 3, line 4 - Move reference to end of sentence

14. pg 3, line 5 - Remove "has"

15. pg 3, line 20 - Change "change" to "changes"

16. pg 3, line 22 - Change "more short" to "shorter"

17. pg 3, line 24 - Change "change" to "changes"

18. pg 4, eq 1 - Add punctuation to equations (including all subsequent equations in paper)

19. pg 4, lines 7-8 - The sentence starting "The population changed..." is not clear.

20. pg 5, line 5 - Change "would" to "could"

21. pg 8, line 11 - Change "which" to "that"

22. pg 8, lines 20-21 - Please write out the equation for the logistic model with noise

23. pg 8, lines 21-22 - Please specify the difference between the "three uncompleted changes". Is the same noise realisation used but for different lengths of trajectories?

24. pg 8, lines 27-29 - The sentence starting with "The results show..." is not clear.

25. pg 10, line 9 - Add "of the largest peak" after "The $k$ value"

26. pg 10, line 10 - Change "are distributed in the" to "also have"

27. pg 10, line 26 - Please quantify what is meant by "tiny"

28. pg 11, lines 1-5 - Percentages of what?

29. pg 11, line 8 - Add "in the PDO" after "abrupt changes"

30. pg 11, line 10 - Add "the" before "two"

31. pg 11, line 13 - Write out the set of sub-sequence lengths in words

32. pg 13, line 1 - Remove "after the abrupt change"

33. pg 13, line 9 - Remove the "a" after each year in the brackets

34. pg 13, line 23 - Remove "the" before "uncompleted" and change "process" to "processes"

35. pg 13, line 24 - Remove "the" before "ideal"

36. pg 13, line 26 - Remove "started"

---

## Author Response (AR1)

**1 # REPLY to RC1**

2 Dear reviewer,

- 3 Thanks for the comments, we modify the manuscript according to the comments and
- 4 reply them one by one as follows. More details are included in the supplement for the
- 5 plain text can not display the entire reply especially the symbols.
- 6

**7 General comments**

- "The paper needs major changes to address all issues. The developments in • 8 9 Subsection 2.1 from page 4 to page 7 line 5 already appear in the first part of section 2 of ref. "Yan PC, Feng GL, Hou W. A novel method for analyzing the 10 11 process of abrupt climate change. Nonlinear Processes in Geophysics 2015; 22:249-258, doi: 10.5194/npg-22-249-2015" pages 250-251 and do not introduce 12 any new information. They should be omitted and cited or resumed." 13 14 REPLY: In the previous paper of ref. "Yan PC, Feng GL, Hou W. A novel method for analyzing the process of abrupt climate change. Nonlinear Processes in Geophysics 15 2015; 22:249-258, doi: 10.5194/npg-22-249-2015", we introduced the detection 16 17 method for transition process of climate change in detail. In this manuscript, we develop this method to predict the uncompleted transition process. Thus, the 18 19 introduction about the method is necessary. Now, we omitted the unnecessary part about the method, and introduce briefly. 20 21 22 • "The method to determine the values of location parameters  $\alpha$  and  $\beta$ , or the 23 position of points A and B is not clearly specified." **REPLY:** We mark point A and point B in figure 2d, and add two lines in figure 2d to 24 explain how to define parameters  $\alpha$  and  $\beta$ . Parameters  $\alpha$  and  $\beta$  are defined to introduce 25 26  $x_a$  and  $x_b$ . 27 "The numerical tests of section 2.1 are not fully specified neither its purpose." 28 ٠ **REPLY:** In section 2.1, we define a new parameter  $\chi$  to simplify Eq.(7) for now, and 29
- 30 the relationship among  $\chi$ ,  $\alpha$ ,  $\beta$  is shown in figure 3b. we find that the changing of  $\chi$  is

| 1  | limited even the change of parameter $\alpha$ is 25%. Besides, there is little influence on the        |
|----|--------------------------------------------------------------------------------------------------------|
| 2  | detection of parameters if the position of the points (the values of parameters $\alpha$ and $\beta$ ) |
| 3  | are indefinite. We add more explanation in the manuscript.                                             |
| 4  |                                                                                                        |
| 5  | • The results of simulated prediction method of section 2.2 are drawn in figure 5                      |
| 6  | but not quantified in the text, so the quality of the prediction method can not be                     |
| 7  | appreciated.                                                                                           |
| 8  | REPLY: We add more description in manuscript as follows:                                        |
| 9  |                                                                                                        |
| 10 | • "For the entire time sequence, there are 500 moments as shown in figure 5a. In                       |
| 11 | figure 5b, only 240 moments are given, and the other moments are unknown.                              |
| 12 | Then, we obtain parameters $v$ and $h$ by regression method. Then, Parameter $u$ is                    |
| 13 | calculated with Eq(8) since parameter $k$ is given. The blue line represent the                        |
| 14 | prediction result. The transition process would be ended in moment 342 with the                        |
| 15 | end state value 2.92. In figure 5c, the end moment and end state are predicted 356                     |
| 16 | and 2.65 respectively when the time sequence is given 250 moments. In figure 5d,                       |
| 17 | the time sequence is given 260 moments, and the end moment and end state are                           |
| 18 | predicted 359, 2.58 respectively. The ideal experiments predict the end moment                         |
| 19 | and end state of transition process successfully. The results also show that the                       |
| 20 | longer the transition process experience, the more accurate the prediction."                           |
| 21 |                                                                                                        |
| 22 | • The method described in the paper is based on the use of continuous functions:                       |
| 23 | piecewise linear functions or logistic model; but it is applied to discontinuous                       |
| 24 | functions: see figures 8 and 11. The lines have jumps and the application of the                       |
| 25 | previous equations to discontinuous functions must be justified.                                       |
| 26 | REPLY: We use the continuous function to express the transition process                         |
| 27 | approximately rather than a piecewise function. And, the real climate time sequence is                 |
| 28 | also continuous indeedly. In the manuscript, we rewrite the paragraph in page 4 line                   |
| 29 | 2-8 as follows:                                                                                        |

| 1  | • "Where t represents time, and $x_t$ represent the system states, which is obtained by                    |
|----|------------------------------------------------------------------------------------------------------------|
| 2  | the linear regression method. It is noted that the climate system is continuous                            |
| 3  | even the sampling sequence that makes it is discontinuous. We used a continuous                            |
| 4  | function to express this transition period approximately, and we also created a                            |
| 5  | novel method to detect the transition period(Yan et al, 2015). The continuous                              |
| 6  | evolution of Logistic model is consistent with the transition process(May, 1976),                          |
| 7  | which is shown in figure 1d. The modified logistic model is expressed as                                   |
| 8  | follows: "                                                                                                 |
| 9  | The table with the results of analysis of past 10 years in Section 3.2 is missing.                         |
| 10 | REPLY: Table 2 was missing, and it is added now.                                                    |
| 11 |                                                                                                            |
| 12 | Specific comments                                                                                          |
| 13 | • 2.1 The detection method of transition process Page 3, lines 21-22. " the                                |
| 14 | period around point $C$ is expanded to a longer period, or the period around point                         |
| 15 | C is observed on a more short time scale " It is not the same. Figure 1b                                   |
| 16 | corresponds to the second option: "observe on a more short time scale" or better                           |
| 17 | "observe on shorter time scale". The idea is that with a more detailed view, the                           |
| 18 | transition process can be observed.                                                                        |
| 19 | REPLY: We did some modification about the description as follows:                                   |
| 20 | "If the period around point C is observed on a more short time scale (as shown in                          |
| 21 | figure 1b), a transition period is obtained, and it is a part of the original time sequence.               |
| 22 | In fact, many abrupt change could be considered to be a transition period with a more                      |
| 23 | detailed view."                                                                                            |
| 24 |                                                                                                            |
| 25 | • In page 6 line 13 and eq. (12), the amplitude of change is denoted by w, but in eq.                      |
| 26 | (13) the notation is changed to $\omega$ .                                                                 |
| 27 | REPLY: This mistake is corrected                                                                    |
| 28 |                                                                                                            |
| 29 | • Page 7, line 6 "According to the numerical experiment". Figure 3 is not a                                |
| 30 | numerical experiment; part (b) is a contour map of $\chi$ for $0 \le \alpha \le 1$ and $0 \le \beta \le 1$ |

and part (a) is a section of that contour map along the line  $\alpha + \beta = 1$  (probably). 1 **REPLY:** This mistake is corrected 2 3 Page 7, line 8. The assertion "the sum of  $\alpha$  and  $\beta$  is 1", does it mean that figure 4 • 3(a) is the profile of figure 3(b) along the diagonal  $\alpha + \beta = 1$ ? Please, clarify. See 5 6 the remark for the caption of Fig. 3. Page 7, lines 13-15. "Let the sum of  $\alpha$  and  $\beta$ be 1, then then the change of parameter  $\chi$  is only related to parameter  $\alpha$ ... (also 7 in figure 3a)". This is obvious,  $\chi$  depends on the two parameters (has two degrees 8 9 of freedom), by imposing a relationship between the two parameters, you reduce the degrees of freedom to one. This sentence does not add any information and 10 11 should be suppressed. REPLY: In fact, due to the symmetry of the transition process to the middle point, we 12 assume that point A and point B are symmetrical about the middle point, and the sum 13 14 of  $\alpha$  and  $\beta$  is 1. We rewrite this part in the manuscript. 15 Page 7, line 16. "In figure 3c, three ideal experiments were carried out...". The 16 • 17 experiments were not carried out in the figure 3c. Figure 3c describes the parameters of the three experiments. As noted before, the experiments deserve 18 19 their own subsection. **REPLY:** These are mistakes. We change "experiment" to be "situation". 20 21 22 • *The setup of the tests is not clearly described. From the figure 3c, we know the* 23 parameters u, v and k, from figure 3c. Other parameters are in table 1. The test setup will be clearer if table 1 would include all parameters for each test. 24 Parameter h0 in table 1 is not defined. Nothing is said about the time span of 25 tests; if the points are randomly perturbed and how. See the previous remark. 26 27 REPLY: In figure3c, we test three different situations with different values of parameters u, v and k. When the values of parameter  $\alpha$  are different, the values of 28 29 parameter  $\chi$  are almost constant. Parameter  $h_0$  and h are defined and obtained as 30 follows in manuscript:

| 1  | "In addition, linear trends of these three ideal models are calculated according to the              |
|----|------------------------------------------------------------------------------------------------------|
| 2  | points and by regression method which are marked as $h_0$ in table 1. The linear trends              |
| 3  | are also calculated by the values of point A and point B with $Eq(5)$ which are marked               |
| 4  | as h in table 1."                                                                                    |
| 5  | Besides, in table 1, the value of parameter $\chi$ is not right. We correct the result in this       |
| 6  | edition.                                                                                             |
| 7  |                                                                                                      |
| 8  | • 2.2 The prediction method of transition process Page 8 line 7, " there is the                      |
| 9  | quartic function relationship between linear trend and amplitude of change." Eq.                     |
| 10 | (13) reads $h=k \omega^2 \chi$ . This equation is quadratic in the amplitude $\omega$ , not quartic. |
| 11 | REPLY: This mistake is corrected                                                              |
| 12 |                                                                                                      |
| 13 | • Page 8, line 18, We are supposing the repetition of events, assuming all events                    |
| 14 | have the same k. We obtain v and h. Which is the value of $\alpha$ ? We are also                     |
| 15 | assuming that $\alpha + \beta = 1$ , so we can calculate $\chi$ .                                    |
| 16 | REPLY: This is a mistake. In section 2.1, an explanation is added. Parameter $\alpha$ is set  |
| 17 | as 0.2, and parameter $\chi$ is 0.2164.                                                              |
| 18 |                                                                                                      |
| 19 | • Page 8, line 23, An ideal time sequence is constructed with a logistic model with                  |
| 20 | parameters $v=-1$ , $u=2$ and $k=0.1$ . But in figure 5, the steady part of the curve is             |
| 21 | well above -1 in the left part and above 2 in the right part. From that graphics,                    |
| 22 | the limits seem to be $v=-1.5$ , $u=2.5$ .                                                           |
| 23 | REPLY: The ideal time sequence is constructed by using the logistic model and                 |
| 24 | random numbers. The random numbers are limited in $(0, 1)$ . We correct this mistake                 |
| 25 | in the manuscript.                                                                                   |
| 26 |                                                                                                      |
| 27 | 3.1 Threshold of stability parameter k                                                               |
| 28 |                                                                                                      |
| 29 | • Page 9, line 15. The origin of the values for the parameter $k$ (green dots) that                  |
| 30 | appear in figure 6a is missing.                                                                      |

| 1  | REPLY: It is true that some green dots in figure 6s are missing. These missing dots |
|----|--------------------------------------------------------------------------------------------|
| 2  | represent the value of parameter $k$ are large. For example, during 1960~1970, the         |
| 3  | threshold of $k$ value is about 200-600 as shown in figure 6a, which means these abrupt    |
| 4  | changes are unstable. Fortunately, most abrupt changes are stable as statistical results   |
| 5  | in figure 7.                                                                               |
| 6  |                                                                                            |
| 7  | • 3.2 Determination of abrupt change and the threshold for initial state v and             |
| 8  | linear trend h                                                                             |
| 9  | Page 11, line 8, "We use the method to analyze" But the method is not specified.           |
| 10 | REPLY: "The method" means the method proposed in section 2.2 in this manuscript.    |
| 11 | We correct this mistake.                                                                   |
| 12 |                                                                                            |
| 13 | • Page 11, line 10, "Parameters v and h are obtained abrupt changes (Table                 |
| 14 | 1)." But the title of table 1 is "The parameters for ideal models" and also written        |
| 15 | in page 7, line 18. Is there another missing table?                                        |
| 16 | REPLY: It should be table 2. Table 2 is added in the manuscript.                    |
| 17 |                                                                                            |
| 18 | • Page 11, line 21. "the abrupt change determined through the percentile                   |
| 19 | threshold method". This method must be described or referenced.                            |
| 20 | REPLY: The percentile threshold method is a statistical method for studying extreme |
| 21 | events. The reference here is added now.                                                   |
| 22 |                                                                                            |
| 23 | • Page 11, line 23. Along the paper, time series were approximated by piece-wise           |
| 24 | continuous functions: the system was in a steady state, and from that value                |
| 25 | changed up or down. But in figure 8 time series are approximated by functions              |
| 26 | with jumps from the value of the steady states to the beginning of the slope lines         |
| 27 | that approximate the changes. These profiles are different from those used in              |
| 28 | figure 5 to simulate the process of recovering the parameters and those of the             |
| 29 | logistic functions.                                                                        |
|    |                                                                                            |

30 **REPLY:** The continuous functions is used to determine the quantitative relationship

| 1  | among parameters. When calculate the values of parameters, we used the optimal                             |
|----|------------------------------------------------------------------------------------------------------------|
| 2  | linear regression method in different segments. Thus, these profiles are linear lines                      |
| 3  | which are not like the curve in figure 5.                                                                  |
| 4  |                                                                                                            |
| 5  | • Page 13, line 4, "the parameter $h=1.054/a$ " The units of h are not clear, what                         |
| 6  | does a mean? Year? The same problem appears in line 9 in the same page.                                    |
| 7  | REPLY: The unit of the PDO index with time(month) is 1, so the linear trend of PDO                  |
| 8  | should be month -1 . We correct this mistake.                                                   |
| 9  |                                                                                                            |
| 10 | • Figure 3 Caption. (a) part: it is not stated which diagonal of (b) refers to, $\alpha = \beta$           |
| 11 | or $\alpha + \beta = 1$ . It, also, would be interesting to mark points 1, 2 and 3 from (b) in             |
| 12 | the part (a) of figure.                                                                                    |
| 13 | REPLY: The diagonal refers to $\alpha + \beta = 1$ . A gray line is added in figure 3(b). Points 1, |
| 14 | 2 and 3 represent three different situations, and we mark them with S1, S2 and S3. We                      |
| 15 | also change the description in table 1.                                                                    |
| 16 |                                                                                                            |
| 17 | Technical corrections                                                                                      |
| 18 | • Cites should be separated from text by a blank, e.g. p. 2 line 1 " change                                |
| 19 | (Charney)" and many more. Page 6 line 4 "Then, we do integration", I                                       |
| 20 | consider better "Then, we integrate"                                                                       |
| 21 | REPLY: We correct these mistakes.                                                                   |
| 22 |                                                                                                            |
| 23 |                                                                                                            |
| 24 |                                                                                                            |

**1 # REPLY to RC2**

2 Dear reviewer,

Thanks for the comments, we modify the manuscript according to the comments and
reply them one by one as follows. More details are included in the supplement for the
plain text can not display the entire reply especially the symbols.

6

**7 General comments**

I. The abstract needs to be made more clear. The phrase "more details of climate change" is too broad and does not explain what exactly is being addressed by the methods presented in the paper. The PDO is also not explained, nor its relation to

11 climate change.

REPLY: We rewrite the abstract as follows, and we add more explanation about thePDO.

14 "Climate change is expressed as a climate system transiting from the initial state to a

15 new state in a short time. The period between the initial state and the new state is

16 defined as transition process, which is the key to connect the two states. By using a

17 piecewise function, the transition process is expressed approximately (Mudelsee,

18 2000). However, the dynamic processes are not included in the piecewise function.

19 Thus, we had proposed a method to study the transition process by using a continuous

20 function. In this manuscript, the method is developed to predict the unfinished

21 transition process based on the dynamic characteristics of the continuous function. We

22 introduce this method in details and apply it to predict end moment and end state of

23 one unfinished transition process of the Pacific Decadal Oscillation (PDO) time

24 sequence, which is a long-lived El Niño-like pattern of Pacific climate variability

25 (Barnett et al, 1999). This method reveals a new relationship during the transition

26 process, which explores a nonlinear relationship between the linear trend and the

27 amplitude (difference) between the initial state and the end state. Since the transition

28 process begins, the initial state and the linear trend are estimated. Then, according to

29 the relationship, the end states and end moment of the unfinished transition process is

30 predicted. The results of either the ideal experiments or the PDO index show good

|   | 1           |   |
|---|-------------|---|
| 1 | nrediction  | ~ |
| 1 | prediction. |   |
|   | 1           |   |

| 3 | • | 2. There is not enough introduction to the methods section before discussing the |
|---|---|----------------------------------------------------------------------------------|
| 4 |   | details of time series analysis.                                                 |

- 5 **REPLY:** In this manuscript, we develop a new method to predict the end state and
- 6 end moment of a uncompleted transition process based on the detection method of
- 7 transition process. The detection method had been published in our previous papers as
- 8 follows. Thus, we rewrite this part about the method in this edition of the manuscript.
- 9 More details about the prediction method are added. Also, we introduce more about
- 10 the ideal experiments in section 2.2.
- 11 Yan PC, Feng GL, Hou W, Wu H Statistical characteristics on decadal abrupt change
- 12 process of time sequence in 500 hPa temperature field. Chinese Journal of
- 13 Atmospheric Sciences 2014; 38 (5): 861–873
- 14 Yan PC, Feng GL, Hou W. A novel method for analyzing the process of abrupt
- 15 climate change. Nonlinear Processes in Geophysics 2015; 22:249-258, doi:
- 16 10.5194/npg-22-249-2015
- 17 Yan PC, Hou W, Feng GL Transition process of abrupt climate change based on
- 18 global sea surface temperature over the past century, Nonlinear Processes in
- 19 Geophysics 2016; 23:115–126, doi:10.5194/npg-23-115-2016
- 20
- 21 *3. The mathematical notation is inconsistent and unclear.*
- 22 Variable k appears to be often interchanged with  $\kappa$
- 23 Parameter k is referred to as both a stability (Fig. 10 and Section 3.3) and instability
- 24 parameter (Fig. 10).
- 25  $\mu$  is never formally introduced and is potentially being exchanged with u.
- 26 **REPLY:** Stability parameter k should be instability parameter. We corrected this
- 27 mistake. The other two mistakes are also corrected.
- 28

30

29 • 4. There is terminology that is used but not defined.

continued process (pg 1, line 12)

- 1 *the filtering process (pg 2, line 16)*
- 2 ramp function (pg 2, line 24)
- 3 *crush (pg 4, line 21)*
- 4 percentile threshold (pg 11, line 21)
- 5 augmented abrupt change (pg 11, line 28)
- 6 **REPLY:** We correct all above mistakes in manuscript one by one. More details are as
- 7 follows:
- 8 continued process (pg 1, line 12)
- 9 The wrong description is moved.
- 10 the filtering process (pg 2, line 16)
- 11 "filtering" is replaced by "transition"
- 12 ramp function (pg 2, line 24)
- 13 The ramp function means piecewise function according to Mudelsee's work. This
- 14 mistake is corrected.
- 15 *crush (pg 4, line 21)*
- 16 It means that the system will be crushed. A reference is added.
- 17 percentile threshold (pg 11, line 21)
- 18 A reference is added.
- 19 augmented abrupt change (pg 11, line 28)
- 20 We rewrite this paragraph, and the mistake is corrected.
- 21
- 22 5. There is inconsistency between Section 2.2 and Section 3. In Section 2.2 it is
- 23 stated that the parameter k cannot be obtained from the data (with no explanation
- 24 as to why), so it is fixed a priori. In Section 3 the parameter k has been estimated
- 25 from a time series, but again with no explanation as to how one would estimate
- 26 *this*.
- 27 **REPLY:** In section 2.2, the ideal time sequence only have one abrupt change, which
- 28 means that we have no way to obtain the parameter k because that there is no more
- 29 other climate changes. While in section 3, the PDO index have several abrupt changes,
- 30 and parameter k is obtained by counting these changes. We rewrite this paragraph in

1 the manuscript as follows.

| 2  | " It has to be noticed that in this ideal time sequence there is just one abrupt change,       |
|----|------------------------------------------------------------------------------------------------|
| 3  | which means that we have no way to obtain the parameter $k$ by counting many                   |
| 4  | changes. Thus parameter $k$ is given directly, and the prediction of the end state             |
| 5  | ( moment) is drawn in figure 5b, 5c and 5d. For the entire ime sequence, there are 500         |
| 6  | moments as shown in figure 5a. In figure 5b, only 240 moments are given, and the               |
| 7  | other moments are unknown. Then, we obtain parameters $v$ and $h$ by regression                |
| 8  | method. Then, Parameter $u$ is calculated with Eq(8) since parameter $k$ is given. The         |
| 9  | blue line represent the prediction result. The transition process would be ended in            |
| 10 | moment 342 with the end state value 2.92. In figure 5c, the end moment and end state           |
| 11 | are predicted 356 and 2.65 respectively when the time sequence is given 250 moments.           |
| 12 | In figure 5d, the time sequence is given 260 moments, and the end moment and end               |
| 13 | state are predicted 359, 2.58 respectively. The ideal experiments predict the end              |
| 14 | moment and end state of transition process successfully. The results also show that the        |
| 15 | longer the transition process experience, the more accurate the prediction."                   |
| 16 |                                                                                                |
| 17 | • 6. In Section 3.1 it is stated "When the length of the sub-sequence is 20 years and          |
| 18 | 30 years, there is only one peak in the distribution of k values: :: " (pg 10, lines           |
| 19 | 21-24). This seems strange, as there are said to be multiple peaks for a smaller               |
| 20 | subsequence (10 years), a single peak for 20 and 30, and then multiple peaks for               |
| 21 | larger subsequences. I would assume there would be a more continuous                           |
| 22 | relationship. This is not discussed why this is not the case. Also, a quantitative             |
| 23 | measure is not specified of what defines a peak.                                               |
| 24 | REPLY : The description in this part was not right. We rewrite this part about the      |
| 25 | values of parameter $k$ . Parameter $k$ characterizes the stability of the system during       |
| 26 | climate change. If it is detected to be large, the system is not stable. The ideal time        |
| 27 | sequences are shown in our previous work as follows. The evolution of the system               |
| 28 | expressed by the logistic model with different stability parameters: (a) the system            |
| 29 | reaches to the stable states with a different initial variable when parameter $k = \pm 0.01$ ; |
| 30 | (b) the system becomes bifurcated when the parameter $k = 105$ ; (c) the system                |
|    |                                                                                                |

- 1 becomes chaotic when the parameter k = 135. However, we can not identify the value
- 2 of parameter directly, but we can find its threshold. Thus, in section 3.1 of this
- 3 manuscript, we obtain parameter k by counting the climate changes of the PDO index.

- 5 By referring: Yan PC, Feng GL, Hou W. A novel method for analyzing the process of
- 6 abrupt climate change. Nonlinear Processes in Geophysics 2015; 22:249-258, doi:

7 10.5194/npg-22-249-2015

8

4

9 • 7. The motivation for Section 3.2 is absent and it is not obvious how this section

- 10 relates to the overall goal of Section 3.
- 11 **REPLY**: In section 3.1, we obtain the parameter *k*, and in section 3.2, we obtain the
- 12 parameters v, h. More explanation as follows in the first paragraph of section 3 is
- 13 added. We also rewrite the first paragraph of section 3.2.
- 14 "During the following research, a transition process starting from 2011 is studied.
- 15 According to the prediction method, several parameters have to be determined in
- 16 advance. We determine parameter k firstly."
- 17
- 18 8. "Abrupt change" appears to be used synonymously with "transition process"
- 19 in Section 3.2 and this does not seem consistent with the rest of the paper. Please
- 20 *maintain the same terminology for clarity.*
- 21 **REPLY**: We check all the manuscript, and change inconsistent description.

| 1  |                                                                                            |
|----|--------------------------------------------------------------------------------------------|
| 2  | • 9. The final paragraph of Section 3.2 (pg 12, lines 10-24) discusses three abrupt        |
| 3  | changes. The previous paragraph discussed four. There is much confusion as to              |
| 4  | what abrupt change events are being discussed throughout this paragraph.                   |
| 5  | REPLY : We correct this mistake and give more explanation in section 3.2.           |
| 6  |                                                                                            |
| 7  | • 10. The lengths of the sub-sequences mentioned in Section 3.2 do not match the           |
| 8  | numbers on the colour bar in Fig 9. It is therefore not clear what Fig 9 is showing.       |
| 9  | REPLY : We add more explanation about figure 9 in section 3.2. In figure 9, the     |
| 10 | transition process starting from 1976 should not be shown. It is corrected now. Only       |
| 11 | the transition process starting from 2007 and 2011 are stated.                             |
| 12 |                                                                                            |
| 13 | • 11. There is no discussion as to which abrupt change detection (year 2007 or             |
| 14 | 2011) is correct, which leads to a lack of motivation for studying only the 2011           |
| 15 | event. It needs to be more clearly explained why the 2011 event is chosen for the          |
| 16 | prediction experiment.                                                                     |
| 17 | REPLY : Both of the climate changes starting from 2007 and 2011 are right. When the |
| 18 | sub-sequence are set as different lengths, which means we test the climate change in       |
| 19 | different time scale, the start moments of climate changes might be different. In this     |
| 20 | manuscript, only the climate change starting from 2011 is studied for testing the          |
| 21 | prediction method. More explanation is added in section 3.                                 |
| 22 |                                                                                            |
| 23 | • 12. The "variation situation of parameter $\mu$ " (pg 13, lines 6-7) was never           |
| 24 | introduced nor explained.                                                                  |
| 25 | REPLY : This is a mistake. It is corrected to be " u ".                      |
| 26 |                                                                                            |
| 27 | • 13. The "prediction result" (pg 13, lines 13-14) was not specified. Additionally, it     |
| 28 | is not clear which prediction is being shown in Fig. 11.                                   |
| 29 | REPLY : We rewrite this paragraph as follows in the manuscript. More explanation    |
| 30 | about the prediction in figure 11 is included.                                             |
|    | 13                                                                                         |

| 1        | "In figure 11, the PDO time sequence is displayed as black line. The period during                                                                                                                                                                         |
|----------|------------------------------------------------------------------------------------------------------------------------------------------------------------------------------------------------------------------------------------------------------------|
| 2        | 2006~2011 is detected as the initial state, and a transition process is increasing from                                                                                                                                                                    |
| 3        | this initial state. It is not able to be known whether the increasing process has been                                                                                                                                                                     |
| 4        | completed or not. Based on the linear regression method, the initial state and the                                                                                                                                                                         |
| 5        | linear trend are obtained and shown as purple dash lines. Then by the method                                                                                                                                                                               |
| 6        | proposed in section 2.2, the end state of transition process are obtained with Eq(8),                                                                                                                                                                      |
| 7        | and it is marked as green dash line Unlike the uncompleted transition process of                                                                                                                                                                           |
| 8        | ideal experiment, the transition process has completed in 2015 since we detected the                                                                                                                                                                       |
| 9        | PDO change in 2016. This transition process started from 2011, and end in 2015. The                                                                                                                                                                        |
| 10       | initial moment and the end moment are marked as black dash lines. However, we are                                                                                                                                                                          |
| 11       | still not sure whether the PDO complete this transition process or not for it it appears                                                                                                                                                                   |
| 12       | at the end of the sequence. As we all know, many statistical methods are not accurate                                                                                                                                                                      |
| 13       | for the detecting both ends of the sequence. Thus, the real PDO sequence during                                                                                                                                                                            |
| 14       | 2016~2017 is added to the end of the PDO time sequence. The PDO value from 2015                                                                                                                                                                            |
| 15       | to 2017 is almost unchanged, which is consistent with the predicted result."                                                                                                                                                                               |
| 16       |                                                                                                                                                                                                                                                            |
| 17       | • 14. Conclusion needs to be expanded upon much more.                                                                                                                                                                                                      |
| 18       | REPLY : We rewrite the conclusion and discussion, and all three following mistakes                                                                                                                                                                  |
| 19       | are corrected now.                                                                                                                                                                                                                                         |
| 20       | The sentence "The abrupt change with smaller time scales has a continuous process,                                                                                                                                                                         |
| 21       | and the abrupt change with larger time scales becomes abrupt change point." (pg $14$ ,                                                                                                                                                                     |
| 22       | lines 8-10) is not easily understandable and alludes to material that does not appear                                                                                                                                                                      |
| 23       | to have been discussed in the paper.                                                                                                                                                                                                                       |
| 24       | The phrase "a detected abrupt change beginning in 2011 appears relatively close to                                                                                                                                                                         |
| 25       |                                                                                                                                                                                                                                                            |
|          | the end of the 115-year sequence, and it is difficult to identify by using other methods"                                                                                                                                                                  |
| 26       | the end of the 115-year sequence, and it is difficult to identify by using other methods"
(pg 14, line 11-13) was not previously discussed in the manuscript. Please expand on                                                                          |
| 26
27 | the end of the 115-year sequence, and it is difficult to identify by using other methods"
(pg 14, line 11-13) was not previously discussed in the manuscript. Please expand on
why the abrupt change is difficult to identify through other methods. |

- 29 increases the possibility of resolving the problem associated with difficult processing
- 30 at the end of a time sequence" (pg 14, lines 14-16). Please add discussion of the

| 1  | problem of difficult processing at the end of a time sequence.              |
|----|-----------------------------------------------------------------------------|
| 2  |                                                                             |
| 3  | Specific comments                                                           |
| 4  | 1. pg 1, line 14 - Change "self" to "itself"                                |
| 5  | REPLY: This mistake is corrected.                                    |
| 6  | 2. pg 1, line 15 - Add full reference to paper on PDO                       |
| 7  | REPLY: The reference is added.                                       |
| 8  | 3. pg 1, line 16 - Remove "And" from beginning of sentence                  |
| 9  | REPLY: This mistake is corrected.                                    |
| 10 | 4. pg 2, lines 4-6 - Add references for each of the fields mentioned        |
| 11 | REPLY: Two references are added.                                     |
| 12 | 5. pg 2, line 6 - Change "famous" to "observed"                             |
| 13 | REPLY: This mistake is corrected.                                    |
| 14 | 6. pg 2, line 8 - Add reference for "Thom's research"                       |
| 15 | REPLY: The reference is added.                                       |
| 16 | 7. pg 2, line 8 - Remove "And" from beginning of sentence                   |
| 17 | REPLY: This mistake is corrected.                                    |
| 18 | 8. pg 2, line 24 - "Ramp Function" does not need to be capitalised          |
| 19 | REPLY: This mistake is corrected.                                    |
| 20 | 9. pg 2, line 26 - "Non-linear Function" and "Ramp Function" do not need to |
| 21 | be capitalised                                                              |
| 22 | REPLY: This mistake is corrected.                                    |
| 23 | 10. pg 2, line 29 - Remove "Besides" from beginning of sentence             |
| 24 | REPLY: This mistake is corrected.                                    |
| 25 | 11. pg 3, line 1 - Change "got reach to" to "reached"                       |
| 26 | REPLY: This mistake is corrected.                                    |
| 27 | 12. pg 3, line 3-4 - Capitalise "decadal oscillation"                       |
| 28 | REPLY: This mistake is corrected.                                    |
| 29 | 13. pg 3, line 4 - Move reference to end of sentence                        |
| 30 | REPLY: The reference is moved                                        |

| 1  | 14. pg 3, line 5 - Remove "has"                                                        |
|----|----------------------------------------------------------------------------------------|
| 2  | REPLY: This mistake is corrected.                                               |
| 3  | 15. pg 3, line 20 - Change "change" to "changes"                                       |
| 4  | REPLY: This mistake is corrected.                                               |
| 5  | 16. pg 3, line 22 - Change "more short" to "shorter"                                   |
| 6  | REPLY: This mistake is corrected.                                               |
| 7  | 17. pg 3, line 24 - Change "change" to "changes"                                       |
| 8  | REPLY: This mistake is corrected.                                               |
| 9  | 18. pg 4, eq 1 - Add punctuation to equations (including all subsequent equations      |
| 10 | in paper)                                                                              |
| 11 | REPLY: This mistake is corrected.                                               |
| 12 | 19. pg 4, lines 7-8 - The sentence starting "The population changed: : : " is not      |
| 13 | clear.                                                                                 |
| 14 | REPLY: This mistake is corrected.                                               |
| 15 | 20. pg 5, line 5 - Change "would" to "could"                                           |
| 16 | REPLY: This mistake is corrected.                                               |
| 17 | 21. pg 8, line 11 - Change "which" to "that"                                           |
| 18 | REPLY: This mistake is corrected.                                               |
| 19 | 22. pg 8, lines 20-21 - Please write out the equation for the logistic model with      |
| 20 | noise                                                                                  |
| 21 | REPLY: The equation is added.                                                   |
| 22 | 23. pg 8, lines 21-22 - Please specify the difference between the "three               |
| 23 | uncompleted changes". Is the same noise realisation used but for different lengths of  |
| 24 | trajectories?                                                                          |
| 25 | REPLY: An entire time sequence with 500 moments is shown in figure 5a and              |
| 26 | three other lengths of time sequences are shown in figures 5b, 5c and 5d respectively. |
| 27 | More explanation is added in title of figure 5.                                        |
| 28 | 24. pg 8, lines 27-29 - The sentence starting with "The results show: : : " is not     |
| 29 | clear.                                                                                 |
| 30 | REPLY: More explanation is added in manuscript.                                 |

| 1  | 25. pg 10, line 9 - Add "of the largest peak" after "The k value"                   |
|----|-------------------------------------------------------------------------------------|
| 2  | REPLY: It is added.                                                          |
| 3  | 26. pg 10, line 10 - Change "are distributed in the" to "also have"                 |
| 4  | REPLY: This mistake is corrected.                                            |
| 5  | 27. pg 10, line 26 - Please quantify what is meant by "tiny"                        |
| 6  | REPLY: We rewrite this sentence as:                                          |
| 7  | It is difficult to detect an abrupt change with huge amplitude if the abrupt change |
| 8  | takes almost no time.                                                               |
| 9  | 28. pg 11, lines 1-5 - Percentages of what?                                         |
| 10 | REPLY: Percentages of all k values. We correct this mistake.          |
| 11 | 29. pg 11, line 8 - Add "in the PDO" after "abrupt changes"                         |
| 12 | REPLY: This mistake is corrected.                                            |
| 13 | 30. pg 11, line 10 - Add "the" before "two"                                         |
| 14 | REPLY: This mistake is corrected.                                            |
| 15 | 31. pg 11, line 13 - Write out the set of sub-sequence lengths in words             |
| 16 | REPLY: We rewrite this sentences in the manuscript.                          |
| 17 | 32. pg 13, line 1 - Remove "after the abrupt change"                                |
| 18 | REPLY: This mistake is corrected.                                            |
| 19 | 33. pg 13, line 9 - Remove the "a" after each year in the brackets                  |
| 20 | REPLY: This mistake is corrected.                                            |
| 21 | 34. pg 13, line 23 - Remove "the" before "uncompleted" and change "process"         |
| 22 | to "processes"                                                                      |
| 23 | REPLY: These mistakes are corrected.                                         |
| 24 | 35. pg 13, line 24 - Remove "the" before "ideal"                                    |
| 25 | REPLY: This mistake is corrected.                                            |
| 26 | 36. pg 13, line 26 - Remove "started                                                |
| 27 | REPLY: This mistake is corrected.                                            |
| 28 |                                                                                     |
|    |                                                                                     |

**A method to predict the uncompleted climate transition process**

| 3                                                                                            | Pengcheng, Yan <mark>1,3, Guolin, Feng2, Wei, Hou2</mark>                                                                                                                                                                                                                                                                                                                                                                                                                                                                                                                                                                                                                                                                                                                                                                                                                                                                                                                                                                                                                                                                                                                                                                                                                                        | 删除: 1                                                                                                                                                                                                                                                                                                                                                                                                      |
|----------------------------------------------------------------------------------------------|-----------------------------------------------------------------------------------------------------------------------------------------------------------------------------------------------------------------------------------------------------------------------------------------------------------------------------------------------------------------------------------------------------------------------------------------------------------------------------------------------------------------------------------------------------------------------------------------------------------------------------------------------------------------------------------------------------------------------------------------------------------------------------------------------------------------------------------------------------------------------------------------------------------------------------------------------------------------------------------------------------------------------------------------------------------------------------------------------------------------------------------------------------------------------------------------------------------------------------------------------------------------------------------------------------------------------------------|------------------------------------------------------------------------------------------------------------------------------------------------------------------------------------------------------------------------------------------------------------------------------------------------------------------------------------------------------------------------------------------------------------|
| 4
5
7
8
9
10                                                                  |  <li>[1] {Institute of Arid Meteorology, China Meteorological Administration, Key
Laboratory of Arid Climatic Change and Reducing Disaster of Gansu Province, Key
Laboratory of Arid Climatic Change and Reducing Disaster of China Meteorological
Administration, China}</li> <li>[2] {National Climate Center, China Meteorological Administration, China}</li> <li>[3] {China Meteorological Administration Training Center, Beijing, China}</li> <li>[*]Correspondence to: Wei Hou (houwei@cma.gov.cn)</li>                                                                                                                                                                                                                                                                                                                                                                                                                                                                                                                                                                                                                                                                                                                                                                                              | 删除: 2
删除: 2
设置格式:左,缩进:首行缩进:0毫米
删除:                                                                                                                                                                                                                                                                                                                                          |
| 11                                                                                           | Abstract                                                                                                                                                                                                                                                                                                                                                                                                                                                                                                                                                                                                                                                                                                                                                                                                                                                                                                                                                                                                                                                                                                                                                                                                                                                                                                                          | 设置格式: 字体: (默认)Calibri, (中文)宋体                                                                                                                                                                                                                                                                                                                                                                              |
| 12

27 | Climate change is expressed as a climate system transiting from the initial state to a new state in a short time. The period between the initial state and the new state is defined as transition process, which is the key part to connect the two states. By using a piece-wise function, the transition process is stated approximately (Mudelsee, 2000). However, the dynamic processes are not included in the piece-wise function. Thus, we had proposed a method to study the transition process by using a continuous function. In this manuscript, this method is developed to predict the uncompleted transition process based on the dynamic characteristics of the continuous function. We introduce this prediction method in details and apply it to three ideal time sequences and the Pacific Decadal Oscillation (PDO), The PDO is a long-lived El Niño-like pattern of Pacific climate variability (Barnett et al, 1999). This method reveals a new quantitative relationship during the transition process, which explores a nonlinear relationship between the linear trend and the amplitude (difference) between the initial state and the linear trend are estimated. Then, according to the relationship, the end state, and end moment of the uncompleted transition process is predicted, | 删除: could be 删除: express 删除: e 删除: predict end moment and end state of one uncompleted transition process of 删除: By considering the short period as a continued process, which is called transition process, more details of climate change would be described according to analysis the time sequence self. We had proposed a method to quantify the transition process of 删除: time sequence, which 删除: s |
|                                                                                              |                                                                                                                                                                                                                                                                                                                                                                                                                                                                                                                                                                                                                                                                                                                                                                                                                                                                                                                                                                                                                                                                                                                                                                                                                                                                                                                                   | 删除: The results of either the ideal experiments or the PDO                                                                                                                                                                                                                                                                                                                                                 |

index show good prediction.

**1 Keywords**

Prediction method; Transition process of abrupt change; System stability; Pacific
 Decadal Oscillation

**4 1. Introduction**

A system transiting from one stable state to another in a short period is called 5 abrupt change (Charney and DeVore, 1979; Lorenz, 1963, 1979). The abrupt change 6 7 system has two or more states (Goldblatt et al, 2006; Alexander et al, 2012), the system swings between these states that are also called attractors in physics. This 8 9 phenomena is verified in many fields including biology (Nozaki, 2001), ecology (Osterkamp et al, 2001), climatology (Thom, 1972; Overpeck and Cole, 2006; Yang et 10 11 al, 2013a, 2013b), brain science (Sherman et al, 1981), etc. The latest observed 12 climate change event is global warning hiatus, which has been studied deeply by many researchers (Amaya et al, 2018; Kosaka and Xie, 2013; Yang et al, 2017). Seven 13 different kind of abrupt changes are mentioned in Thom's research(1972). Over the 14 15 last several decades, many methods have been proposed to identify different kinds of abrupt change (Li et al, 1996), like Moving T-Test, Cramer's (Wei, 1999), 16 Mann-Kendall (MK, Goossens and Berger, 1986), Fisher (Cabezas and Fath, 2002), 17 etc. It is noticed that most abrupt change detection methods suggests that the abrupt 18 change is around a turning point. The significant difference between the average 19 20 values of the two sequences on both two sides of the turning point is defined as the index to measure the abrupt change. This kind of detection method has a drawback. It 21 is difficult to detect the abrupt change occurs at the end of sequence. 22 23 Mudelsee (2000) studied the abrupt change of a time sequence and illustrated that abrupt change has a duration, which can be quantitatively described with a 24 piece-wise (ramp) function, We, developed the detection method by using a 25

- 26 continuous function to replace, the ramp function( Yan et al, 2014, 2015). The new
- 27 method can confine the beginning and ending points of abrupt change and

删除: and global sea surface temperature system. And the quantitative relationships among the parameters characterizing the abrupt changes is revealed during the transition process. In this paper, we develop this method to predict the end moment (state) if the transition process has not been completed.

**删除: is**

删除:, human behavior 删除: famous 删除: And o 删除: only 删除:, and the 删除: a 删除: the 删除: Notably, when the

删除: Notably, when the abrupt change occurs at the end of sequence, i

删除: . The filteringtransition process considers the continuity of the time series and provides a very important technique for the detection and prediction of abrupt change. For an abrupt climate change event, the determination of the start moment and end moment for the abrupt change is helpful for better ....

删除: namely, the transitional process. Moreover, it

删除: the

删除: R

删除: F

删除: , which is more advanced than traditional metrics of abrupt change

- 删除: Yan et al. (2014, 2015)
- 删除: with a Nnon-linear Function replacing
- 删除: R
- 删除: F

quantitatively describes the process of abrupt climate change, and three parameters 1 are introduced. A quantitative relationship among the parameters is revealed (Yan et al, 2 2015). The relationship could be used to predict the end moment (state) if the system 3 had left the original state but not yet reached to the new state, which is defined as 4 5 uncompleted transition process. 6 In this manuscript, three ideal time sequences are tested to study the prediction method. The prediction method is also applied to study the climate transition process 7 of the PDO, which is an important signal, that reveals climatic variability on the 8 9 decadal timescale (Mantua et al, 1997; Barnett et al, 1999; Zhang et al, 1997; Yang et al, 2004). Previous studies (Lu et al, 2013; Trenberth and Hurrell, 1994) have 10 11 indicated that there are many climate changes in the PDO over the past 100 years. Most researches mentioned the climate changes happened in the 1940s and 1970s. 12 During the 1940s, the PDO transited from a high state to a low state, while during the 13 14 1970s, it did the opposite. All this changes and their processes had been studied in our previous researches (Yan et al, 2015, 2016). The climate transition processes were 15 explored clearly. However, we still can not know when the transition processes finish 16 17 its increasing or decreasing to a stable state if the transition process has begun. We develop a new method to predict the end state and the end moment of a transition 18 19 process based on the quantitative relationship.

**20 **2. Methods**

**21 **2.1** The detection method of transition process**

The real time sequence changes abruptly as shown in figure 1a, and the system jumps to a high state in point C. If the period around point C is observed on a shorter time scale (as shown figure 1b), a transition period is obtained, and it is a part of the original time sequence. In fact, many abrupt changes could be considered to be a transition period with a more detailed view, The transition period was expressed with an ramp function in Mudelsee's research (2000), as shown in figure 1c, and the time

3

删除· 删除: paper 删除: several 删除: hen, t 删除: abrupt 删除: change 删除: Pacific dDecadal oOscillation ( 删除:) 删除: (Mantua et al, 1997; Zhang et al, 1997) 删除: . The PDO has not only affect inter-decadal change .... 删除: Mantua et al. 1997; 删除: have been 删除: oscillations 删除:, and in particular, the most famous are the ... 删除: transformed 删除: and 删除: the 删除:;Yan et al, 删除: research on the prediction of abrupt change process .... 删除: is expanded to a longer period, or the period around  $\overline{\cdots}$ 删除: more 删除: in figure 1b 删除: which 删除: sh 删除: much more than a pointwhen the observed window .... 删除: Mudelsee(2000) indicated that this kind of abrupt .... 删除: can be 删除: ( 删除:.

删除: Besides, a

删除: got

删除: A

sequence is divided into three segments, including two equilibrium states and one 1

increasing state. The ramp function is as follows: 2

3
$$x_t = \begin{cases} x_1 & t \le t_1 \\ x_1 + (t - t_1)(x_2 - x_1)/(t_2 - t_1) & t_1 \le t \le t_2 \\ x_2 & t \ge t_2 \end{cases}$$
 (1)

Where t represents time, and  $x_t$  represent the system states, which is obtained by 4 5 the linear regression method. It is noted that the climate system is continuous even the sampling sequence that makes it is discontinuous. We used a continuous function to 6 express this transition period approximately, and we also created, a novel method, to 7 8 detect the transition period (Yan et al, 2015). Here, the detection method is troduced 9 briefly, The continuous evolution of Logistic model is consistent with the transition process, (May, 1976), which is shown in figure 1d. The modified logistic model is 10 11 expressed as follows: (2)

$$\qquad \dot{x} = k(x-u)(v-x)$$

Parameters u and v represent the two equilibrium states respectively. Parameter k13 represents the switching between different states, and it is defined as instability 14 15 parameter. As shown in figure 2a, parameters u and v being fixed, and setting k as 0.5, the system transiting to the new state costs a shorter time than that setting k as 0.4. If 16 parameter k is set large enough, the system collapses and becomes chaotic ( as shown 17 in figure 2b). When parameter k is set to different values, more situations have been 18 discussed in detail in the previous research (Yan et al, 2016). The result shows that 19 20 parameter k characterizes the stability of the system (the larger the absolute value, the more unstable the system). According to Thom's theory (1972), the system described 21 by a quadratic function would exhibit tipping-point abrupt change, which the system 22 23 jumps from one state to a new state abruptly. Thus, we did some mathematical derivation to Eq. (2), and the general potential energy is obtained as follows: 24

4

$$V_{(x)} = -\int_0^x \ddot{x} dx = -\int_0^x 2k^2 [x - (u + v)/2] (x - u)(x - v) dx$$

=  $\frac{k^2}{2} [x^4 - 2(u + v)x^3 + (u^2 + v^2 + 4uv)x^2 - 2(u + v)uvx]$

删除: hen, t 删除: following 删除: wa 删除: used to fit the transition period. 设置格式: 字体: 非倾斜 删除: and  $x_t$  represents its state, 设置格式: 字体: 非倾斜, 非上标/ 下标 删除: . Based on 删除: fitting 删除:  $x_i$  was obtained to describe the system's behavior 删除: By referring this work, 删除: (Yan et al, 2015) 删除: was proposed 删除: a brief introduction to 删除: as follows 删除: by using the logistic model 删除: . The logistic model was created to study the population of Humans or insects. The population changed( increased or decreased) looks like the expanded of abrupt change, whic .... 删除: c

删除: and its image is shown in figure 1d

删除: speed of

删除:

[revised manuscript text omitted]

删除: According to Thom's(1972) theory, the system 删除:4 删除: which 删除:, and the system is able to transit between them 设置格式: 字体: 倾斜 设置格式: 字体: 倾斜 删除: ••• 删除: ) in 删除:6 删除: And, 删除: Besides, t 设置格式: 字体: 倾斜 删除: with 删除:5  $h = \sum_{i=n_1+1}^{n_1+n_2} \overline{i} \cdot \overline{x}_i / \sum_{i=n_1+1}^{n_1+n_2} \overline{i}^2$ (( ... 删除: 删除:7 删除: And b 设置格式: 字体: 倾斜 设置格式: 字体: 倾斜 设置格式: 左 删除: Then, we do integration for both sides of Eq.(2). In  $\overline{\cdots}$ 设置格式: 字体: 倾斜 删除:10 删除: and Eq.(11) are introduced into Eq.(7), and the ••• 删除: part,  $(\beta - \alpha) / \ln((\beta(\alpha - 1))/(\alpha(\beta - 1)))$ 设置格式: 字体: 非倾斜 删除:, 删除: and the relationship among  $\chi$ ,  $\alpha$ ,  $\beta$  is displayed in figure 1. 删除: 12 删除:13

删除:::

$$\qquad \frac{h = \frac{x_b - x_a}{\left(\mu - \nu\right)\kappa} \ln \frac{x_0 - \mu}{x_0 - \nu} \left(\frac{x_b - \nu}{x_b - u} - \frac{x_a - \nu}{x_a - u}\right)}{= \kappa (\mu - \nu)^2 \frac{(\beta - \alpha)}{\ln \frac{\beta(\alpha - 1)}{\alpha(\beta - 1)}}}$$

3  $h = \kappa \omega^2 \chi$

In order to determine the value of parameter  $\chi$ , the relationship among  $\chi$ ,  $\alpha$ ,  $\beta$  is 4 5 displayed in figure 3b, The dash line in figure 3a is the profile of the diagonal in figure 3b, which represents that the sum of  $\alpha$  and  $\beta$  is 1. Parameter  $\chi$  changes little 6 7 when the location parameter varies in a certain range as marked with warm color in figure 3b. It means that the closer the points (A and B) are to the middle point, the 8 9 more significant the linear feature is. Then, the process between point A and point B10 can represent the whole transition process as shown in figure 3c. It is noted that the transition process is symmetrical about the middle point approximately. Thus, we 11 12 assume that point A and point B are symmetrical about the middle point, and the sum 13 of  $\alpha$  and  $\beta$  is 1. The change of parameter  $\chi$  is only related to parameter  $\alpha$  (or parameter  $\beta$ ), as shown in the diagonals in figure 3b (also in figure 3a). Parameter  $\gamma$  changes 14 little when parameter  $\alpha$  is about 0.2 or larger. In figure 3c, three different situations 15 are carried out to study the influence of parameter  $\alpha$  on parameter  $\chi$ . In each situation, 16 points (A and B) are set to be different positions, and their parameters were calculated 17 18 respectively in table 1. The parameters  $\alpha$  are set as 0.20, 0.25, 0.15 respectively in three different situations marked with S1, S2 and S3. For S2 and S3, both of the 19 percentages of  $\alpha$  changing to S1 are 25%, while the percentages of  $\chi$  changing are 20 only 5.15% and 6.76% respectively, which means the percentage change of  $\chi$  is much 21 less than  $\alpha$ . In addition, linear trends of these three ideal models are calculated 22 according to the points and by regression method which are marked as he in table 1. 23 The linear trends are also calculated by the values of point A and point B with Eq(5) 24 which are marked as h in table 1. It is noted that although the positions of points are 25

6

删除: According to the numerical experiment, the relatior .... 删除:, 删除: and f 删除: indicate 删除: is obvious 删除: turning points ( 删除:) 删除: To simplify the parameters, 设置格式: 字体: 倾斜 设置格式: 字体: 倾斜 删除: Let 删除: be 删除:,then t 删除: 删除: three ideal experiments 删除: we 删除: test 删除: experiment 删除: we 删除: 删除: as 删除: t 删除: / 删除:/ 删除: test 删除: test 删除: test 删除: test 删除: 删除: 4.07 删除: 7.73 设置格式: 字体: 倾斜

设置格式: 字体: 倾斜, 下标

**删除: 13**

\_ (7)

(8)

1 different, the trend obtained according to the points is almost the same as that 2 obtained by regression method. The error percentages are 2.36%, 2.25%, 1.38% 3 respectively, which means that when the position of the points (the values of 4 parameters  $\alpha$  and  $\beta$ ) are indefinite, there is little influence on the detection of 5 parameter h. Thus, in the following sections parameter  $\alpha$  is set as 0.2, and parameter  $\chi$ 6 is 0.2164

**7 **2.2** The prediction method of transition process**

8 Eq. (8) shows the quantitative relationship among linear trend, instability 9 parameter, and amplitude of change. There is a linear relationship between linear trend and, instability parameter; and there is the quadratic function relationship 10 between linear trend and amplitude of change. We did reveal this quantitative 11 relationship much more than in theory but in real time series, (Yan et al, 2016). Based 12 13 on this relationship, we are going to create a new method to deal with the problem that the transition process has not finished. During the real time sequence, the system 14 transits away from the original state, but it has not reached to a new state as shown in 15 16 figure 4. The red line represents the period which has been experienced, while the 17 gray line represents the period which hasn't been experienced. Based on the system states which is far away from the original state, a quasi linear extension of the 18 19 transition process is established (dash line). Then the parameters v and h are obtained by Eq. (4). Assuming that the parameter k satisfies the statistics in the history of the 20 system, the parameter u can be predicted by Eq. (8), and the end moment is also 21 22 predicted apparently.

23
$$\begin{cases} x_{t} = x_{t-1} + kt(x_{t} - u)(v - x_{t}) \\ x'_{t} = x_{t} + random_{t} \end{cases}$$

As shown in figure 5, four ideal time sequences are constructed by using the logistic model and random numbers as Eq. (9). An entire time sequence with 500 moments is shown in figure 5a and three other lengths of time sequences are shown in

figures 5b, 5c and 5d respectively. The parameters v, u and k of the logistic model are

删除: changes slightly 设置格式: 字体: 倾斜 删除: s

删除:13 删除: stability parameter 删除: stability parameter 删除: by studying the sea surface temperature 删除: improve 删除: which 删除: i 删除: been in 删除: the method of 删除: 5) and Eq. (6 删除: on the basis of 删除:13 删除· of this transition process 删除: An ideal time sequence is constructed by using the logistic model and random numbers to achieve the prediction as 设置格式:字体: (中文) 宋体, 小四, 非升高量/降低量 删除: a 删除:, and t 删除: uncompleted 删除: chang 删除: es

(9)

| 1  | set as -1.0, 2.0, 0.1, for the ideal time sequence, and the random number is limited in          |
|----|--------------------------------------------------------------------------------------------------|
| 2  | 0-1. The parameters $v$ , $h$ are obtained by regression method before making prediction. |
| 3  | It has to be noticed that in this ideal time sequence there is just one abrupt change,           |
| 4  | which means that we have no way to obtain the value of the parameter $k$ by counting             |
| 5  | many changes, Thus parameter $k$ is given directly, and the prediction of the end state          |
| 6  | ( moment) is drawn in figure 5b, 5c and 5d. For the entire time sequence, there are              |
| 7  | 500 moments as shown in figure 5a. In figure 5b, only 240 moments are given, and                 |
| 8  | the other moments are unknown. Then, we obtain parameters $v$ and $h$ by regression              |
| 9  | method. The parameter $\mu$ is calculated with Eq. (8). The blue line represent the              |
| 10 | prediction result. The transition process would be ended in moment 342 with the end              |
| 11 | state value 2.92. In figure 5c, the end moment and end state are predicted to be 356             |
| 12 | and 2.65 respectively when the time sequence is given 250 moments. In figure 5d, the             |
| 13 | time sequence is given 260 moments, The end moment and end state are predicted to                |
| 14 | be 359, and 2.58 respectively, The end moment and the end state of prediction result             |
| 15 | match the presetting lines. The results also show that the longer the transition process         |
| 16 | experience, the more accurate the prediction,                                                    |
|    |                                                                                                  |
| 17 | 3. Results                                                                                |
| 10 | In order to test the validity of this prediction method in a real alimete system we              |
| 18 | In order to test the validity of this prediction method in a real climate system, we             |
| 19 | apply this method to predict the uncompleted transition process of the PDO, The PDO,             |
| 20 | index data used is from website of the University of Washington                                  |
| 21 | (http://research.jisao.washington.edu/pdo/), The time period from January of 1900 to             |

[revised manuscript text omitted]

删除: star points) is consistent with the prediction results

| 1 | 设置格式: 字体: 倾斜                                                          |
|---|-----------------------------------------------------------------------|
| 1 | 设置格式: 字体: 倾斜                                                          |
| 1 | 设置格式: 字体: 倾斜                                                          |
| Ì | 设置格式: 字体: 倾斜                                                          |
| Ì | 删除: In this paper,                                                    |
| Ì | 删除: the                                                               |
| Ì | 删除: the                                                               |
| 1 | 删除: we                                                                |
| 1 | 删除: The quantitative relationship between the parameter $\overline{}$ |
| 1 | 删除: the                                                               |
|   | 删除: abrupt                                                            |
| Ì | 删除: started                                                           |
| Ì | 删除:,                                                                  |
| Ì | 删除: and t                                                             |
| Ì | 删除:,                                                                  |
|   | 删除: ri                                                                |
|   | 删除: s                                                                 |
|   | 删除: . The study indicates that the PDO system over the $\fbox$        |

- 1 only by time sequence. It is noted that the uncompleted climate change we studied is
- 2 closed to the end of the sequence. Due to the lake of enough data, it is difficult to
- 3 study the end of time sequence by using other statistical methods,

**4 Acknowledgements**

- 5 We thank two anonymous reviewers for their valuable suggestions. This study
- 6 was jointly sponsored by National Key Research and Development Program of China
- 7 (Grant No. 2018YFE0109600), National Natural Science Foundation of China (Grant
- 8 Nos. 41875096, 41775078, 41675092), Meteorological scientific research project of
- 9 Gansu Meteorological Bureau (MS201914),

**10 References**

[revised manuscript text omitted]

设置格式:段落间距段前:0.5 行,段后:0.5 行

删除:

•

| I |         | Situations                | ~             |                            | k0           | k                               | 160 h /h                         |          |
|---|---------|---------------------------|---------------|----------------------------|--------------|---------------------------------|----------------------------------|----------|
|   |         | Situations                | a             | χ                          | no           | n                               | n 0- n  / n |          |
|   |         | S 1                | 0.20          | 21 .64 E-2          | 12.99E-4     | 12.69E-4                        | 2.36%                            | 删除: test |
|   |         | \$ 2               | 0.25          | 22.76E-2                   | 9.10E-4      | 8.90E-4                         | 2.25%                            | 删除: 87   |
|   |         | S 3                | 0.15          | 20.18E-2                   | 32.27E-4     | 32.72E-4                        | 1.38%                            | 删除: test |
| 2 | Table2. | Parameters v and h        | obtained w    | th different sub-se        | quences      |                                 |                                  | 删除: test |
|   |         | Length of
sub-sequence | s
( | tart moment
year.month) | v     | h (month -1 ) |                                  |          |
|   |         | 10a                |               | 2011.06             | -0.45 | 1.054                    |                                  |          |
|   |         | 20a                |               | 2011.06             | -0.03 | 1.054                    |                                  |          |

0.36

0.41

0.227

0.227

2007.11

2007.11

1 Table 1. The parameters of ideal models

30a

40a

---

## Referee Report (RR1)

**Review of revised manuscript "A method to predict the uncompleted climate transition process"**

There are still some major changes that need to be made before I feel this paper is suitable to be considered for publication. This first issues I will note are related to some of my previous referee comments that were not properly addressed in the new version. I will list these and expand on them below. I also include additional general and specific comments for the revised manuscript.

**Original comments not addressed**

1. **General comments #2** *There is not enough introduction to the methods section before discussing the details of time series analysis.*

    - The authors responded to this by rewriting some parts of the Subsections 2.1 and 2.2. I still feel as though a few sentences of introduction to this general Section 2 are necessary before there is a jump to Subsection 2.1.

2. **General comments #3** *Variable k appears to be often interchanged with $\kappa$*

    - Figure 7 still uses $\kappa$ on the x-axis.

3. **General comments #6** *In Section 3.1 it is stated "When the length of the sub-sequence is 20 years and 30 years, there is only one peak in the distribution of k values . . . " This seems strange, as there are said to be multiple peaks for a smaller subsequence (10 years), a single peak for 20 and 30, and then multiple peaks for larger subsequences. I would assume there would be a more continuous relationship. This is not discussed why this is not the case. Also, a quantitative measure is not specified of what defines a peak.*

    - The reply from the authors does not address my comments at all. The authors discuss the stability behaviour of $k$ rather than the behaviour of the distribution for different sub-sequence lengths. They mention that the text around this discussion in the manuscript is changed when it has not been. Additionally they still do not specify how they define a peak.

4. **General comments #8** *"Abrupt change" appears to be used synonymously with "transition process" in Section 3.2 and this does not seem consistent with the rest of the paper. Please maintain the same terminology for clarity.*

    - The phrase was changed in many places to read "transition change". This is redundant. I have noted all the instances below (in specific comments) where it needs to be fixed to "transition process", along with a few instances of "abrupt change" that were missed.

5. **General comments #10** *The lengths of the subsequences mentioned in Section 3.2 do not match the numbers on the colour bar in Fig 9. It is therefore not clear what Fig 9 is showing.*

    - The authors' response does not address the colour bar mismatch at all. The labels on the colour bar still do not clearly represent years from 10 to 60 in intervals of 1.

**Additional general comments on revised manuscript**

1. The statements "According to Thom's theory ... the general potential energy is obtained as follows" (pg 4, lines 21-24) are not clear. What is meant by "the system described be a quadratic function" and how is it related to the general potential energy? Do you mean that the potential energy should be described by a quadratic function?

2. Equations 5 and 6 need to be incorporated into sentences and full punctuation is needed for all equations. Additionally, $h$ is defined twice. If I understand the rest of the section correctly I believe one should be $h_0$.

3. The introduction of location parameters $\alpha, \beta$ (pg 5, line 14) seems out of place. They are not used in any equations up to that point. Please explain more formally how these are related to the system states $x_a$ and $x_b$ in Eq. 5?

4. Equations 7 and 8 should be incorporated into the sentences where they are introduced for improved clarity and understanding for the reader.

5. What does the term "indefinite" mean here (pg 7, line 4)? It seems to be used synonymously with "unknown".

6. In Eq. 9 the random terms are just labeled as "$random_t$". It would be more appropriate to label with a variable name (e.g. $\sigma$, $\eta$, etc.) and state from which distribution the random variable is chosen.

7. The sentence "Therefore, for the entire sub-sequence, there are many transition changes" (pg 9, line 19) is unclear what the message. In particular the phrase "the entire sub-sequence" is used multiple times throughout the paper and I am not sure what is meant by it.

**Specific comments/technical corrections**

1. pg 1, lines 16-17 - Add reference to previous paper in sentence "Thus, we had proposed ... " .

2. pg 1, line 25 - "Since" → "as"

3. pg 2, line 14 - "kind" → "kinds"

4. pg 2, line 16 - "like" → "such as"

5. pg 2, line 18 - "suggests" → "suggest"

6. pg 2, line 22 - "change occurs" → "change that occurs"

7. pg 3, lines 4-5 - "as uncompleted" → "as an uncompleted"

8. pg 3, line 11 - "climate" → "abrupt"

9. pg 3, line 14 - "this" → "of these"

10. pg 3, line 15 - "researches" → "research"

11. pg 3, line 17 - "its" → "their"

12. pg 3, line 27 - "an" → "a"

13. pg 4, line 4 - "Where" should be lowercase and no indent

14. pg 4, line 5 - "continuous even" → "continuous; it is"

15. pg 4, line 5 - "troduced" → "introduced"

16. pg 4, line 9 - "of Logistic" → "of the logistic"

17. pg 4, eq 2 - "," → "."

18. pg 4, line 14 - "as instability" → "as the instability"

19. pg 4, line 22 - "change which" → "change in which"

20. pg 4, eq 3 - "," → "."

21. pg 5, line 6 - "by regression" → "by the regression"

22. pg 5, line 8 - "of second" → "of the second"

23. pg 5, line 16 - "." → ":"

24. pg 5, eq 6 - "," → "."

25. pg 7, line 17 - "hasn't" → "has not"

26. pg 7, line 18 - "is" → "are"

27. pg 7, line 22 - Remove "apparently"

28. pg 8, line 3 - "noticed" → "noted"

29. pg 8, line 9 - "represent" → "represents"

30. pg 8, line 14 - "of prediction" → "of the prediction"

31. pg 8, lines 25-26 - "We determine parameter $k$ firstly" → "We first determine parameter $k$"

32. pg 9, lines 7-8 - "transition changes of the PDO mainly" → "main transitions of the PDO"

33. pg 9, line 9 - "this two transition changes" → "these two transitions"

34. pg 9, line 10 - Remove "the" before $k < 0$

35. pg 9, lines 16-17 - "The dots in the left side region are more than the dots" → "There are more dots in the left side region than"

36. pg 9, line 20 - "transition change" → "transition process"

37. pg 9, line 21 - "begins" → "begin"

38. pg 9, line 22 - "transition changes" → "transition processes"

39. pg 9, line 23 - "transition changes" → "transition processes"

40. pg 9, line 25 - "abrupt change processes" → "transition processes"

41. pg 9, line 27 - "abrupt changes" → "transition processes"

42. pg 10, line 4 - "there are two kind of transitions" → "the two transitions"

43. pg 10, line 23 - "transition changes" → "transition processes"

44. pg 10, line 25 - "transition changes" → "transition processes"

45. pg 10, line 26 - "stated" → "shown"

46. pg 10, line 26 - "transition change" → "transition process"

47. pg 10, line 28 - "when the transition change" → "for the transition processes"

48. pg 11, line 1 - "20 years or 30 years" should be 10 and 20 to be consistent with the figure

49. pg 11, line 1 - "transition change" → "transition process"

50. pg 11, lines 4-5 - "transition change" → "transition process"

51. pg 11, lines 5-6 - "-0.45 and -0.03, respectively" - Respectively of what? Not clear to what the two values refer. If it is the two choices of subsequence please make that clear.

52. pg 11, line 7 - "transition change" → "transition process"

53. pg 11, lines 9-10 - "Why does the length . . . ?" - Should not use a question here.

54. pg 11, line 11 - "transition change" → "transition process"

55. pg 11, line 12 - "transition change" → "transition process"

56. pg 11, line 13 - What is meant by "anyway"?

57. pg 11, line 13 - "change" → "transition process"

58. pg 11, lines 14-15 - The sentence "When the sub-sequence is set to be 10 years . . ." is unnecessary. The 10-year sub-sequence was already previously discussed.

59. pg 11, line 17 - "transition change experiences a period" → "transition occurs over a period of time"

60. pg 11, line 17 - "as" → "the"

61. pg 11, lines 18-19 - "looks like the increasing part" → "appears to be increasing"

62. pg 11, line 19 - "transition change" → "transition processes"

63. pg 11, line 20 - "sub-sequence one year by one year" → "the sub-sequences to yearly intervals"

64. pg 11, line 21 - "and" → "up to"

65. pg 11, line 22 - "transition changes" → "transition processes"

66. pg 11, line 22 - "figure 9, the" → "figure 9, when the"

67. pg 11, line 23 - "about" → "approximately"

68. pg 11, line 23 - "transition changes" → "transition processes"

69. pg 11, line 25 - "transition change" → "transition process"

70. pg 11, line 26 - "abrupt change" → "transition process"

71. pg 11, line 27 - "sub-sequence" → "sub-sequences"

72. pg 11, line 28 - "transition change" → "transition process"

73. pg 11, line 29 - "sub-sequence" → "sub-sequences"

74. pg 12, line 1 - What is meant by "threshold"? Should it be parameter $v$?

75. pg 12, line 3 - "transition change" → "transition process"

76. pg 12, line 4 - "transition change" → "transition process"

77. pg 12, line 8 - "transition change" → "transition process"

78. pg 12, line 9 - "transition change" → "transition process"

79. pg 12, line 11 - "transition change" → "transition process"

80. pg 12, lines 19-20 - "to explain how the prediction method works briefly" → "to briefly explain how the prediction method works"

81. pg 12, line 20 - "as black" → "as a black"

82. pg 12, line 26 - "as green" → "as a green"

83. pg 13, line 2 - Remove comma after 2011

84. pg 13, line 2 - "end" → "ends"

85. pg 13, line 5 - Remove "As we all know"

86. pg 13, line 2 - "this" → "these"

87. pg 13, line 20 - "transition change began" → "transition process beginning"

88. pg 13, line 24 - "Accord" → "According"

89. pg 13, line 25 - "been happened" → "occurred"

90. pg 13, line 25 - "parameters. Then, we" → "parameters and"

91. pg 13 & 14, lines 26-28 & 1 - The statements "However, if the transition process has not begun . . . only by time sequence." are unnecessary.

92. pg 21, line 7 - "panel" → "Y-axis"

93. pg 21, line 8 - "panel" → "Y-axis"

94. pg 21, line 9 - "transition changes" → "transition processes"

95. pg 21, line 12 - "abrupt change" → "the transition processes"

96. pg 22, line 5 - "sub-sequence" → "sub-sequences"

97. pg 22, line 9 - "is" → "represent the"

98. pg 22, line 9 - "sequences" → "sequence"

99. pg 22, line 10 - "of transition change" → "of the transition process"

100. pg 22, line 11 - "line" → "lines"

101. pg 23, line 2 - "transition changes" → "transition processes"

102. pg 24, line 3 - I am not sure what is meant by "the line with starts is the PDO index after 2015"

103. pg 24, lines 3-4 - "start moment" - should this be start and end moment?

---

## Referee Report (RR2)

**General comments**

The paper has been improved but some issues remain.

The method of obtaining the value of $k$ remains unclear. Page 8, line 27 "… we can get the value of parameter $k$ by counting all changes …". If it is obtained by counting changes, how can be negative?

The use of functions with jumps (see fig. 8) remains unjustified. All the developments and definitions are done with the logistic model or piecewise **continuous** functions, but the application to the real system has jumps between the initial state and the transition process and between the transition process and the final state.

**Specific comments**

Page 7, line 26. It is stated "*The parameters v, u and k of the logistic model are set as -1.0, 2.0, 0.1, ...*" but in figure 5, the $v$ and $u$ values seem to be -0.5 and 2.5 respectively. In page 8, the recovered values are 2.92, 2.65 and 2.58, converging to 2.5 as deduced from the graphic instead to the value stated in the text (2.0).
The recovered values show big differences (2.92 – 46% error, 2.65 – 32.5% error and 2.58 – 29% error) with respect to to the original value (2.0). This lack of agreement contrasts with the good results in the real case showed in fig. 11.
May be, this disagreement due to the introduction of random variations (uniformly distributed?) which are always positive (range 0-1).

**Technical corrections**

Page 2, line 19-20 "It is difficult to detect the abrupt change occurs at the end of sequence." Is there something missing? For example "… abrupt change [that] occurs …".

Page 4, line 6 "detect the transition period (Yan et al, 2015). Here, the detection method is troduced" Should be "introduced"? Misspelling?

Page 13, line 26 "Due to the lake of enough data…" Misspelling? Lack?

---

## Referee Report (RR3)

**Review of revised manuscript "A method to predict the uncompleted climate transition process"**

The manuscript in this state continues to be unsuitable for consideration of publication. Much of the grammar is unclear and the scientific findings of the paper get lost due to this. There also continue to be comments not addressed from previous reviews and more issues with the continuity and clarity of the paper in the revised version. Below I will simply list the main issues of the paper, noting which ones are still persisting from previous versions. In this review I will not go into the technical corrections and leave it to the authors to fix the numerous grammar mistakes.

**General comments**

- **Section 2.1**

  1. It is never explicitly stated which equilibrium states are represented by $u$ and $v$ (which is the start and which is the end state).

  2. Figure 2a shows an example for $k = -0.4$, not $k = 0.4$. Adjust figure or text accordingly.

  3. Absolute value of $k$ had not been previously discussed, which may also relate to my previous comment. Maybe discussing this earlier clears up what is going on in the figure.

  4. **(previous comment)** The statement "*According to Thom's theory (1972), the system described by a quartic function ...*" is still unclear. The authors' response cleared up the confusion between quartic and quadratic from the previous version, but did not make this sentence clearer to the reader. According to the theory, if the system's *general potential energy* is described by a quartic function then the system has a tipping point. The system itself does not need to be described by a quartic function.

  5. Variable $n_2$ is introduced in the text, but $n_1$ and $n_3$ are not.

  6. **(previous comment)** The parameter $h$ is defined twice, where one is an approximation of the other. The parameters should be labelled differently to clarify which $h$ one is discussing in the rest of the manuscript.

  7. **(previous comment)** Punctuation was added to Eq. 5 and 6, but they were not properly incorporated into a sentence as suggested.

  8. The variables $x_0$ and $t_0$ are used in Eq. 6 and never introduced in the text.

  9. **(previous comment)** Punctuation was added to Eq. 5 and 6, but they were not properly incorporated into a sentence as suggested.

10. **(previous comment)** The relation of $\alpha$ and $\beta$ to $x_a$ and $x_b$ was explained in the authors' response to my comments but was not clarified in the text of the paper. The mathematical relationship is only seen in Figure 2d but it should also be in the text as it is necessary for the understanding of the mathematical derivation in this section.

11. **(previous comment)** The parameter $\mu$ is used in equation 7 but not introduced previously.

12. **(previous comment)** Eq. 7 and 8 are still not clearly integrated into the text as previously suggested. The equations should not be referenced before they are introduced. (This point holds for all equations in the manuscript.)

13. Parameter $\omega$ is never defined in the text but is used in Eq. 8.

- **Section 2.2**

  1. Eq. 9 seems out of place and not incorporated into any text where it is introduced.

  2. The statement "*The end moment and the end state of the prediction result match the presetting lines.*" is not entirely accurate. The two predictions using 250 and 260 moments can be argued to match the truth, but the first using 240 moments appears to obviously overestimate the end state.

- **Section 3.1**

  1. **(previous comment)** The phrase "transition change" (or "transition changes") is still used in this section and subsequent sections.

  2. The text states that Figure 6b has a variation of 20-60 years of sub-sequence lengths, yet the figure appears to show from 15-60 years.

  3. **(previous comment)** When Figure 7 is first discussed in the text, it is unclear which sub-sequence results are being discussed. I assume its the 10-year sub-sequence, but it is never. specified. Also, a quantitative definition of a peak is never introduced in the text, nor in the authors' previous response (the authors' mention "extremely high frequency" without an actual threshold on what defines a frequency to be "extremely high".

- **Section 3.2**

  1. **(previous comment)** The phrase "transition change" (or "transition changes") is still used in this section.

- **Section 4**

  1. Along with my comment #2 for Section 2.2, not all of the ideal experiments accurately predict the end state. This first (using 240 moments) seems to overestimate the state. This should be noted and discussed.

2. **(previous comment)** The phrase "transition change" is still used in this section.

- **Figures**

  1. In Figure 2a the legend appears to be wrong. The lines do not correspond to their starting (v) and ending (u) values

  2. Caption of Figure 6 is wrong. The X-axis of 6b shows sub-sequence length in years, not months.

  3. Caption of Figure 7 has not been adjusted to reflect the new figure.

---

## Referee Report (RR4)

**Referee report on the paper "A method to predict the uncompleted climate transition process" by Pengcheng Yan, Guolin Feng, and Wei Hou.**

**General comment:**

The authors developed a method to predict the uncompleted transition process based on the dynamic characteristics of the continuous function instead of a piecewise one, applying it to three ideal time sequences and the Pacific Decadal Oscillation (PDO). The manuscript is of potential interest for the broad community of NPG, however in its present form is difficult to read, some concepts and findings are unclear and also some previous methods should be used as a comparison. Thus, I recommend to revise the manuscript based on the suggestions given below.

**Major comments:**

1. According to the Thom's theory (1972) there are at least 4 different types of dynamics which describe different physical systems. In page 5, line 2 the authors state "It means that Eq. (2) describes a system with tipping-point abrupt change". This is not correct since the tipping-point picture proposed by Thom (1972) is associated with a thirth-order polynomial function, generally known as "fold catastrophe". The authors refer to a quartic polynomial function which is related to the so-called "cusp catastrophe", that can be related to different kinds of bifurcation (e.g., the pitchfork). The bistable function the authors found is indeed very common for the climate system (as for simple energy-balance models, see a lot of papers by Ghil and co-authors, or the famous stochastic resonance mechanism, see Benzi et al. papers) but the equation governing the dynamics of the system should be characterized by a third-order polynomial in the righthand term. Indeed, given a forcing F(x) the dynamics of the system can be described by dx = F(x) dt and assuming that F(x) is a polynomial function of order n, then V(x) is a polynomial function of order n+1. By looking at Eqs. (1) and (2) of the present paper there is a discrepancy between the quadratic forcing term of Eq. (1) and the quartic potential function of Eq. (2). The authors need to carefully consider this point.

2. Fig. 6: in my opinion the authors should revise this figure, also concering comments raised by previous reviewers. There is still a discrepancy between the different form of k, it is not simply readable, and it seems to me that there is something wrong in plotting the green dots. Please revise.

3. Fig. 7: as for Fig. 6 in my opinion the authors should revise this figure. Where is the gray region in the upper-right corner stated in the caption? Moreover, the authors should also insert

4. I'm not sure I completely understood the main purposes of the manuscript as written in the present form. Indeed, it seems to me that the authors state that "By using a piece-wise function, the transition process is stated approximately" (lines 14-15 in the Abstract) and that "Thus, we had proposed a method (Yan et al, 2015, 2016) to study the transition process by using a continuous function" (lines 17-18 in the Abstract) but they are using a piecewise prediction method which is based on a linearization of the logistic equation. So, what would be the main benefit? Why not to directly use the logistic fit for investigating the transition? Moreover, the authors use the term "tipping point" but they are not considering/discussing implications of tipping points as well as comparing their results with established methods for tipping point evaluation (variance, autocorrelation, ..., see papers by Ditlevsen, Lenton, and colleagues). I urge the authors reorganize both the introduction and the conclusion sections to take into account these aspects.

**Minor comments:**

- Page 2, line 6: the term "attractors" is too wide in this context, probably it is better to use "fixed points" or "equilibrium states".

- Page 4, line 9: the term "discontinuous" seems not to be appropriate, should be better "discrete".

- Page 4, line 19: "k as 0.4" should be "k as 0.5". Indeed, in Fig. 1 being u and v fixed then k values are

chosen as 0.3 and 0.5, when k=-0.4 is chosen different values for u and v are considered.

- Fig. 5: are the authors showing xt' and not xt? Please confirm.

- Check some English forms through the text.

---

## Author Response (AR2)

[revised manuscript text omitted]

Dear reviewers,

We do thank you very much. we revise this manuscript based on your comments and reply them one by one as follows.

**REPLY TO RC1**

**Original comments not addressed**

1. General comments #2 There is not enough introduction to the methods section before discussing the details of time series analysis.

• The authors responded to this by rewriting some parts of the Subsections

2.1 and 2.2. I still feel as though a few sentences of introduction to this general

Section 2 are necessary before there is a jump to Sub-section 2.1.

**REPLY:** It seems that we misunderstood the original comment. Before Section 2.1, we added some explanations as follows:

*It is necessary to describe the transition process quantitatively before the*

*prediction of the uncompleted climate transition process. We had proposed a detection*

*method by using the logistic model to obtain a transition process. In section 2.1, the*

*method is introduced briefly. On the basis of the detection method, the prediction*

*method for studying the uncompleted transition process is further developed in section*

*2.2.*

2. General comments #3 Variable k appears to be often interchanged with κ

• Figure 7 still uses κ on the x-axis.

**REPLY:** Figure 7 is replaced with a new edition, and "*κ*" is changed to be "*k*".

3. General comments #6 In Section 3.1 it is stated "When the length of the sub-sequence is 20 years and 30 years, there is only one peak in the distribution of k values . . . " This seems strange, as there are said to be multiple peaks for a smaller sub-sequence (10 years), a single peak for 20

and 30, and then multiple peaks for larger sub-sequences. I would assume there would be a more continuous relationship. This is not discussed why this is not the case. Also, a quantitative measure is not specified of what defines a peak.

• The reply from the authors does not address my comments at all. The authors discuss the stability behaviour of $k$ rather than the behaviour of the distribution for different sub-sequence lengths. They mention that the text around this discussion in the manuscript is changed when it has not been. Additionally they still do not specify how they define a peak.

**REPLY:** In section 3.1, figure 7 is replaced to be a new edition as follows. We consider the extremely high frequency marked by blue circle in the following figure as peaks. It is true that there is a continuous relationship. There is only one main peak for sub-sequence of 10a, 20a, and 30a. There are two main peaks for sub-sequence of 40a, 50a, and 60a. We modify the description in the manuscript as follows:

[Figure]

*When the length of the sub-sequence is 20 years and 30 years, there is only one main peak in the distribution of k values, and the parameter k value of the peak is about 0,*

*which means that the transition change is more stable than the other situations. When*

*the length of the sub-sequence is 40, 50, or 60 years, there are two main peaks.*

4. General comments #8 "Abrupt change" appears to be used synonymously with "transition process" in Section 3.2 and this does not seem consistent with the rest of the paper. Please maintain the same terminology for clarity.

   • The phrase was changed in many places to read "transition change". This is redundant. I have noted all the instances below (in specific comments) where it needs to be fixed to "transition process", along with a few instances of "abrupt change" that were missed.

**REPLY:** Thank you very much for these very detailed comments. We have corrected all the mistakes according to the "*Specific comments/technical corrections*".

5. General comments #10 The lengths of the sub-sequences mentioned in Section 3.2 do not match the numbers on the colour bar in Fig 9. It is therefore not clear what Fig 9 is showing.

   • The authors' response does not address the colour bar mismatch at all. The labels on the colour bar still do not clearly represent years from 10 to 60 in intervals of 1.

**REPLY:** It seems that we misunderstood the mismatch about the color bar. We replace figure 9 with a new edition as follows.

[Figure]

**Additional general comments on revised manuscript**

1. The statements "According to Thom's theory . . . the general potential energy is obtained as follows" (pg 4, lines 21-24) are not clear. What is meant by "the system described be a quadratic function" and how is it related to the general potential energy?

Do you mean that the potential energy should be described by a quadratic function?

**REPLY:** It should be "quartic function". In part C, section 5.3 of Thom's book

" *Stability Structural and Morphogenesis*", he introduced a general potential energy,

$V = \dfrac{1}{4}x^4 + \dfrac{1}{2}x^2 + vx$ . The quartic function describes a system with tipping point, which is an abrupt change type. Thus, we study the general potential energy of Eq.2 in the manuscript as:

$$V_{(x)} = -\int_0^x \ddot{x}\,dx = -\int_0^x 2k^2 [x - (u+v)/2](x-u)(x-v)\,dx$$
$$= \frac{k^2}{2}[x^4 - 2(u+v)x^3 + (u^2 + v^2 + 4uv)x^2 - 2(u+v)uvx]$$

, which is similar to Thom's equation. It also means that the equation, $\dot{x} = k(x-u)(v-x)$, describes a system with tipping point abrupt change. We change "quadratic function" to be "quartic function" in section 2.1.

2. Equations 5 and 6 need to be incorporated into sentences and full punctuation is needed for all equations. Additionally, $h$ is defined twice. If I understand the rest of the section correctly I believe one should be $h0$.

**REPLY:** The full punctuations are added after equations 5 and 6. In equation 5, we define $h$ with two points ($A$ and $B$). In equation 7, we calculate $h$ value by using the solution of function (2), and we have the quantitative relationship. For the relationship (Eq. 8), $h = k(u-v)^2 \chi$ , $h$,$u$,$v$ are obtained by equation 4. Then, we have $k$ value.

3. The introduction of location parameters $\alpha$, $\beta$ (pg 5, line 14) seems out of place.

They are not used in any equations up to that point. Please explain more formally how these are related to the system states $x_a$ and $x_b$ in Eq. 5?

**REPLY:** The parameters $\alpha$ and $\beta$ are introduced to describe the positions of points $A$

and $B$. Figure 2d is shown as follows, and the transition process is extracted and placed on the right. The parameters $\alpha$ and $\beta$ are defined with $u, v$ and $x$. In figure 3, a numerical test is stated to study the impact of the positions of points $A$ and $B$ (also parameters $\alpha$ and $\beta$) on parameter $\chi$. Finally, it is noted that parameter $\chi$ change sightly and it is given as an invariant constant.

[Figure]

4. Equations 7 and 8 should be incorporated into the sentences where they are introduced for improved clarity and understanding for the reader.

**REPLY:** Punctuation after equation 8 has been modified.

5. What does the term "indefinite" mean here (pg 7, line 4)? It seems to be used synonymously with "unknown".

**REPLY:** When point $A$ and point $B$ change around their original positions ($\alpha$=0.2 and $\beta$=0.8), the $\chi$ value changes very little. This means that we don't have to know the exactly positions of point $A$ and $B$. we can approximate the value of $\chi$. We rewrite this part in the manuscript.

6. In Eq. 9 the random terms are just labeled as "random$_t$". It would be more appropriate to label with a variable name (e.g. $\sigma$, $\eta$, etc.) and state from which distribution the random variable is chosen.

**REPLY:** We modify Eq. 9 and replace $random_t$ with $\eta_t$. More description is added as follows:

*As shown in figure 5, four ideal time sequences are constructed by using the logistic model and random numbers as Eq. (9), where $\eta_t$ represents the random number.*

7. The sentence "Therefore, for the entire sub-sequence, there are many transition changes" (pg 9, line 19) is unclear what the message. In particular the phrase "the entire sub-sequence" is used multiple times throughout the paper and I am not sure what is meant by it.

**REPLY:** Three "entire sub-sequence" were included. They are not necessary, and all of them are removed. In section 3.1, the sentence " Therefore, for the entire sub-sequence, there are many transition changes" has been changed to be:

*More transition processes are detected.*

**Specific comments/technical corrections**

All technical mistakes are corrected based on the comments. They are too many to be listed. We sincerely thank these detailed comments.

**REPLY TO RC2**

**General comments**

The paper has been improved but some issues remain.

The method of obtaining the value of k remains unclear. Page 8, line 27 "… we can get the value of parameter k by counting all changes …". If it is obtained by counting changes, how can be negative?

**REPLY:** The "changes" should be "abrupt changes". According the logistic model, there is no abrupt change if $k = 0$. If $k \neq 0$, the abrupt change occurs. We can calculate the $k$ value based on the abrupt change. If $k > 0$, the time sequence transits from the negative phase to the positive phase and vice versa.

The use of functions with jumps (see fig. 8) remains unjustified. All the developments and definitions are done with the logistic model or piecewise continuous functions, but the application to the real system has jumps between the initial state and the transition process and between the transition process and the final state.

**REPLY:** In order to get optimums fitting effect, we did not use continuous piece-wise function to fit the real system. In most cases, due to the continuity of the real time series itself, the fitting results have a slight jump, which has little impact on the final prediction results. If there is a significant jump in the time series, the prediction results will be significantly affected. In the future, we will carry out more ideal experiments to study the influence of the abnormal jump of the sequence on the prediction results.

**Specific comments**

Page 7, line 26. It is stated "The parameters v, u and k of the logistic model are set as -1.0, 2.0, 0.1, ..." but in figure 5, the v and u values seem to be -0.5 and 2.5 respectively.

**REPLY:** The parameters $v$, $u$ and $k$ of the logistic model are set as -1.0, 2.0, 0.1, for the ideal time sequence, and the random number is limited in 0-1. We built the ideal time sequence by using the sum of the logistic model and the random number. Thus, the $v$ and $u$ values seem to be -0.5 and 2.5 respectively in figure 5.

In page 8, the recovered values are 2.92, 2.65 and 2.58, converging to 2.5 as deduced from the graphic instead to the value stated in the text (2.0). The recovered values show big differences (2.92 – 46% error, 2.65 – 32.5% error and 2.58 – 29% error) with respect to to the original value (2.0). This lack of agreement contrasts with the good results in the real case showed in fig. 11. May be, this disagreement due to the introduction of random variations (uniformly distributed?) which are always positive (range 0-1).

**REPLY:** Due to the introduction of random variations (white noise), the value of end state is 2.5 according to the ideal time sequence which is built by using the sum of the logistic model (the value of $u$ is 2.0) and the random number (the average value is 0.5). The recovered values are 2.92, 2.65 and 2.58 when lengths of time sequence are set to be 240, 250 and 260 respectively. Then, the deviation rate are ( (2.92-2.50)/2.50= ) 16.8%, ( (2.65-2.50)/2.50= ) 6% and ( (2.58-2.50)/2.50= ) 3.2%. The prediction value of $u$ is approaching to be 2.5 when the length of time sequence is given enough to cover the entire transition process.

**Technical corrections**

Page 2, line 19-20 "It is difficult to detect the abrupt change occurs at the end of sequence." Is there something missing? For example "… abrupt change [that] occurs …".

**REPLY:** This mistake is corrected.

Page 4, line 6 "detect the transition period (Yan et al, 2015). Here, the detection method is troduced" Should be "introduced"? Misspelling?

**REPLY:** This mistake is corrected.

Page 13, line 26 "Due to the lake of enough data…" Misspelling? Lack?

**REPLY:** This mistake is corrected.

---

## Author Response (AR3)

Dear editors and reviewers,

Thanks a lot for the comments and suggestions about our research.

Over the last month, we tried our best to fix this manuscript and reply the comments one by one as follows. The useful suggestions help us understand the key and advantages of our prediction method deeply. The key is the instability parameter $k$, which is also proposed in our method for the first time. At present, there is no other way to detect the transition process at the end of the time sequence, which is actually related to extreme events. Therefore, we hope that the transition process which has just started can be detected and predicted. That is what we did in this manuscript.

Best wishes,

All authors.

================================================================

Reviewer's Comment 1

General comments

• Section 2.1

1. It is never explicitly stated which equilibrium states are represented by $u$ and $v$

(which is the start and which is the end state.

**[REPLY]:** Below equation (2), we added "Parameters $u$ and $v$ represent the two equilibrium states respectively. Parameter $u$ represents initial state, and parameter $v$

represents end state. " to state the meaning of parameters $u$ and $v$.

2. Figure 2a shows an example for $k = -0.4$, not $k = 0.4$. Adjust figure or text accordingly.

**[REPLY]:** We check the parameters values, and they should be:

$v$=1.0, $u$=2.0, $k$=-0.4 for the gray line,

$v$=-1.0, $u$=1.0, $k$=0.4 for the gray line,

$v$=-1.0, $u$=1.0, $k$=0.3 for the gray line.

We correct the text and figure 2a as follows:

*As shown in figure 2a, parameters u and v being fixed, and setting k as 0.4 ( the dash*

*gray line), the system increasing to the new state costs a shorter time than that setting*

*k as 0.3 (the black line). It is noted that if k<0 ( as v=1.0, u=2.0 and k=-0.4 ), the*

*system decreases from state 2.0 to state 1.0 as the gray line. This states that if the*

*absolute value of k is relatively large, the shorter the transition time of the system,*
*that is, the more unstable(Yan et al, 2016). If parameter k is set large enough, the*
*system collapses and becomes chaotic as shown in figure 2b.*

[Figure]

*Figure 2. (a)The transition processes of system swinging between different stable*
*states since the parameters are different.*

3. Absolute value of *k* had not been previously discussed, which may also relate to my
previous comment. Maybe discussing this earlier clears up what is going on in the
figure.
**[REPLY]:** As in the previous reply, we added a discussion about parameter *k*. This
result is based on a numerical experiment on the parameter *k*, which is discussed in
detail in the previous article (Yan et al, 2016) as follows:
*" ⋯and the parameter κ is shown in Fig. 2. In Fig. 2a, the system reached the same*
*state (x=μ) during the evolution by changing the initial variable in threshold ($x_0 \in$ (v,*
*μ)), when parameter κ (κ=0.1) was positive (black lines). The dashed lines show the*
*condition that the system reached in the state x = v, when the parameter κ (κ < 0) was*
*negative. In Fig. 2b, the parameter κ. (κ = 105) was larger than before, and the*
*system became bifurcated. If the parameter κ. (κ = 135) is much larger, the system*
*will become chaotic as shown in Fig. 2c. Therefore, the parameter κ is a stability*
*parameter, and the parameters μ and v are the start and end states before and after*
*the abrupt change⋯"*

[Figure]

*Figure 2. The evolution of the system over time, with different stability parameters: (a)*
*the system reaches to the stable states with a different initial variable when parameter*
*κ = ±0.01; (b) the system becomes bifurcated when the parameter κ = 105; (c) the*
*system becomes chaotic when the parameter κ = 135. [Reference from the previous*
*article (Yan et al, 2016)]*

4. (previous comment) The statement "*According to Thom's theory (1972), the system described by a quartic function . . . "* is still unclear. The authors' response cleared up the confusion between quartic and quadratic from the previous version, but did not make this sentence clearer to the reader. According to the theory, if the system's general potential energy is described by a quartic function then the system has a tipping point. The system itself does not need to be described by a quartic function.

[REPLY]: According to Thom's theory, the quartic function of generalized potential energy characterizes an abrupt change. The characteristic of equation (2) has to be confirmed by checking its generalized potential energy. The expression of external force, which is connected with the acceleration, is the second derivative of the state variable. Then, the external force f(x) is obtained as follows:

$$f(x) = \frac{d^2 x}{dt^2} = \frac{d[k(x-u)(v-x)]}{dt} = \frac{d(-kx^2 + k(u+v)x - kuv]}{dt}$$

$$= [-2kx + k(v+u)]\frac{dx}{dt} = 2k^2[x - (u+v)/2](x-u)(x-v)$$

$$= k^2[2x^3 - 2(u+v)x^2 + (u^2 + v^2 + 4uv)x - (u+v)uv]$$ .

Then, its generalized potential energy can be expressed as:

$$V(x) = -\int_0^x f(x)dx = -\int_0^x k^2[2x^3 - 2(u+v)x^2 + (u^2 + v^2 + 4uv)x - (u+v)uv]dx$$

$$= \frac{k^2}{2}[x^4 - 2(u+v)x^3 + (u^2 + v^2 + 4uv)x^2 - 2(u+v)uvx]$$ , which means that the logistic model, $\frac{dx}{dt} = k(x-u)(v-x)$, can be used to represent the abrupt change.

[REPLY]: Variables $n_1, n_2, n_3$ represent the lengths of the first, the second and the third segment respectively. This part has been corrected.

[REPLY]: Actually, the linear trend parameter *h* was only defined to be the ratio of system state change to time change. The value of parameter *h* has different way to obtain. One is to calculate by regression method, and another is to obtain according to the definition, which by using the solution of the equation.
- The regression method to calculate *h* based on equation (6) in the manuscript:

$$h = \sum_{i=n_1+1}^{n_1+n_2} \bar{i} \cdot \bar{x}_i \Big/ \sum_{i=n_1+1}^{n_1+n_2} \bar{i}^2$$ .

- Another way to calculate *h* by using the solution,

$t = \dfrac{1}{k(u-v)}\ln(\dfrac{x-v}{x-u}\cdot\dfrac{x_0-u}{x_0-v}) + t_0$, of equation (2),

$h = \dfrac{x_b - x_a}{t_b - t_a} = \dfrac{(\beta - \alpha)(u - v)}{\dfrac{1}{k(u-v)}\left(\ln(\dfrac{x_b-v}{x_b-u}) - \ln(\dfrac{x_a-v}{x_a-u})\right)} = k(u-v)^2\,\dfrac{(\beta-\alpha)}{\ln\dfrac{\beta(\alpha-1)}{\alpha(\beta-1)}} = k(u-v)^2\,\chi$

                                                   .

This is the key part of the prediction method we proposed in this manuscript. We can
not get the values of parameter $k$ directly. Only by using the above equation can
calculate $k$ value on the basis of obtaining the other values of parameters $h$, $v$, $u$, where
the $\chi$ value is constant. Then, the threshold of parameter $k$ is revealed.

7. (previous comment) Punctuation was added to Eq. 5 and 6, but they were not
properly incorporated into a sentence as suggested.
[REPLY]: Thanks a lot for such detailed suggestion, and we correct these mistakes.

8. The variables $x_0$ and $t_0$ are used in Eq. 6 and never introduced in the text.
[REPLY]: The variables $t_0$ and $x_0$ represent the initial time and the initial state
respectively. They are mentioned in the manuscript.

9. (previous comment) Punctuation was added to Eq. 5 and 6, but they were not
properly incorporated into a sentence as suggested.
[REPLY]: We correct the mistakes.

10. (previous comment) The relation of $\alpha$ and $\beta$ to $x_a$ and $x_b$ was explained in the
authors' response to my comments but was not clarified in the text of the paper. The
mathematical relationship is only seen in Figure 2d but it should also be in the text as
it is necessary for the understanding of the mathematical derivation in this section.
[REPLY]: We supplement more equations now, and introduce how to get them step by
step.

11. (previous comment) The parameter $\mu$ is used in equation 7 but not introduced
previously.
[REPLY]: The parameter $\mu$ should be $u$, which is modified in the text.

12. (previous comment) Eq. 7 and 8 are still not clearly integrated into the text as
previously suggested. The equations should not be referenced before they are
introduced. (This point holds for all equations in the manuscript.)
[REPLY]: We apologize that we did not explain them clearly in the previous
responses. Now, more equations are introduced to explain how to work about the
prediction method step by step.

13. Parameter $\omega$ is never defined in the text but is used in Eq. 8.

[REPLY]: The parameter $\omega$ is defined to be the difference between the initial state $v$ and the end state $u$, which is called the amplitude of change. Now, we delete the parameter $\omega$, and describe the amplitude of change with $u$-$v$ directly in text.

• Section 2.2

1. Eq. 9 seems out of place and not incorporated into any text where it is introduced.

[REPLY]: We used the equation to build the ideal time sequence. The description is corrected as follows:

*In order to test the prediction method,an ideal time sequence is constructed by using equation (12), which is the sum of the logistic model and random numbers, where $\eta_t$ represents the random number,*

$$\begin{cases} x_t = x_{t-1} + kt(x_t - u)(v - x_t) \\ x'_t = x_t + \eta_t \end{cases} \qquad (12)$$

2. The statement "*The end moment and the end state of the prediction result match the presetting lines.*" is not entirely accurate. The two predictions using 250 and 260 moments can be argued to match the truth, but the first using 240 moments appears to obviously overestimate the end state.

[REPLY]: We add calculation about the prediction errors for the three time sequences, and they are 0.37, 0.27 and 0.26 respectively, which state that the prediction error is small when the transition process experienced is longer. We give more discussion about the prediction error as follows in the text:

*"The prediction end moment and the prediction end state are basically consistent with the original time sequences. However, the average absolute prediction errors of the three time sequences are 0.37, 0.27 and 0.26 respectively. When the length of the sequence is 240, the prediction state is overestimated, and the average absolute prediction error is 0.37. With the length of the system experienced expanding, the prediction error decreases. The prediction states are very close to the original states when the length is 260. Therefore, in the actual prediction, we hope that the transition process has been experienced for a long enough time, which will help to predict accurately."*

• Section 3.1

1. (previous comment) The phrase "*transition change*" (or "*transition changes*") is still used in this section and subsequent sections.

[REPLY]: The mistake is corrected.

2. The text states that Figure 6b has a variation of 20-60 years of subsequence lengths, yet the figure appears to show from 15-60 years.

[REPLY]: The mistake is corrected to be 15-60 years in the text.

3. (previous comment) When Figure 7 is first discussed in the text, it is unclear which sub-sequence results are being discussed. I assume its the 10-year sub-sequence, but it is never specified. Also, a quantitative definition of a peak is never introduced in the text, nor in the authors' previous response (the authors' mention "*extremely high frequency*" without an actual threshold on what defines a frequency to be "*extremely high*".

[REPLY]: Thanks a lot for the comment about how to quantify the peak, which is also confuse us. Now, the reference line is set as 5%. Only the frequency which is bigger than 5% can be considered to be one mainly peak. Besides, we also recalculate the frequencies of $k$ values, the values in the previous figure 7 (a) were truly wrong. There is one mainly peak for that the length of sub-sequence is less than or equal to 30 years, and there are two mainly peaks for that the length of sub-sequence is greater than 30 years. **There is a "continuous relationship" as mentioned in the previous GENERAL COMMENT #6.** Without this comment, the mistake about the $k$ values for that the length is 10 years would not be found. Therefore, we are very grateful to the reviewers for insisting on this comment.

More discussion is supplemented in text as follows:

*"The small figure in figure 7 (a) shows that the k values (marked with green dots) are more that 100 during 1960~1970 when the length of sub-sequence is 20 years. The frequencies of parameter k values when the length of sub-sequence are 10, 20, 20, 40, 50, and 60 years respectively are displayed in figure 7. It is noted that some of the values of parameter k are so large that their frequencies are almost zero which are not necessary to be counted. The frequencies of k values which belongs to -10 to 10 are shown in figure 7. By considering the frequency which is bigger than 5% as a peak, there is only one peak when the length of sub-sequence is 10 years. Most of the k values are concentrated around zero, which means that the transition processes detected are stable. For the situations which the lengths of sub-sequences are 20 and 30 years, there is one mainly peak, and it is near zero. When the lengths of sub-sequences are more than 30 years, there are two mainly peaks. One is near zero, and another is much less than zero. It states that the much more unstable transition processes are detected when the lengths of sub-sequences are large. From the perspective of the k threshold values, the k values in the range of (-10, 10) accounts for 63.90%, 70.64%, 77.00%, 90.05%, 93.69%, and 89.90% of all k values for that the lengths of sub-sequences are 10, 20, 30, 40, 50, 60 years respectively. They are 55.64%, 67.52%, 73.99%, 83.45%, 85.46%, 84.82% for the range of (-5, 5) and 35.64%, 62.22%, 59.36%, 68.28%, 66.25%, 47.55% for the range of (-2, 2). In the following studies, the k values are mainly considered to be in the range of (-2, 2)."*

• Section 3.2

1. (previous comment) The phrase "*transition change*" (or "*transition changes*") is still used in this section.

[REPLY]: This mistake is corrected.

• Section 4

1. Along with my comment #2 for Section 2.2, not all of the ideal experiments accurately predict the end state. This first (using 240 moments) seems to overestimate the state. This should be noted and discussed.

[REPLY]: We discuss more about the overestimate when the length is 240, and the prediction error is 0.37. With the length of the system experienced expanding, the prediction error decreases. The prediction states are very close to the original states when the length is 260. This reveals that we will predict the turning point (end state and end moment) well if the transition process has been experienced for a long period.

2. (previous comment) The phrase "*transition change*" is still used in this section.

[REPLY]: This mistake is corrected.

• Figures

1. In Figure 2a the legend appears to be wrong. The lines do not correspond to their starting (*v*) and ending (*u*) values

[REPLY]: The mistakes are corrected for the legend of figure 2a as the reply to comment #2 for Section 2.1 in page 1.

2. Caption of Figure 6 is wrong. The X-axis of 6b shows sub-sequence length in years, not months.

[REPLY]: This mistake is modified.

3. Caption of Figure 7 has not been adjusted to reflect the new figure.

[REPLY]: The caption is changed to be "Figure 7. Statistical results of instability parameters for different sub-sequences lengths. The X-axis is the value of parameter k, and the Y-axis is the statistical frequency for the sub-sequence length of 10a, 20a 30a 40a 50a and 60a".
* * *
Reviewer's Comment 2

Major comments:

1. According to the Thom's theory (1972) there are at least 4 different types of dynamics which describe different physical systems. In page 5, line 2 the authors state "It means that Eq. (2) describes a system with tipping-point abrupt change". This is not correct since the tipping-point picture proposed by Thom (1972) is associated with a thirth-order polynomial function, generally known as "fold catastrophe". The authors refer to a quartic polynomial function which is related to the so-called "cusp catastrophe", that can be related to different kinds of bifurcation (e.g., the pitchfork). The bistable function the authors found is indeed very common for the climate system (as for simple energy-balance models, see a lot of papers by Ghil and coauthors, or the famous stochastic resonance mechanism, see Benzi et al. papers) but the equation governing the dynamics of the system should be characterized by a third-order polynomial in the righthand term. Indeed, given a forcing $F(x)$ the dynamics of the system can be described by $dx = F(x) dt$ and assuming that $F(x)$ is conservative then there exists a potential function $V(x)$ such that $F(x) = -dV(x)/dx$. This means that if $F(x)$ is a polynomial function of order n, then $V(x)$ is a polynomial function of order n+1. By looking at Eqs.(1) and (2) of the present paper there is a discrepancy between
the quadratic forcing term of Eq. (1) and the quartic potential function of Eq. (2). The
authors need to carefully consider this point.
[REPLY]: Thanks a lot for the suggestion. The logistic model represents the first
derivation of the state variable with respect to time,

$$\frac{dx}{dt} = k(x-u)(v-x)$$ .

The second derivative with respect to time is the acceleration, which is proportional to
the external forcing marked with $f(x)$. Then, we have

$$f(x) = \frac{d^2x}{dt^2} = \frac{d[k(x-u)(v-x)]}{dt} = \frac{d(-kx^2 + k(u+v)x - kuv]}{dt}$$

$$= [-2kx + k(v+u)]\frac{dx}{dt} = 2k^2[x-(u+v)/2](x-u)(x-v)$$

$$= k^2[2x^3 - 2(u+v)x^2 + (u^2+v^2+4uv)x - (u+v)uv]$$ .

The above formula was omitted from the original manuscript, now we supplement it
in the current version. The generalized potential energy (Benzi et al, 1982) is
expressed as the integral of the external force to the state,

$$V(x) = -\int_0^x f(x)dx = -\int_0^x k^2[2x^3 - 2(u+v)x^2 + (u^2+v^2+4uv)x - (u+v)uv]dx$$

$$= \frac{k^2}{2}[x^4 - 2(u+v)x^3 + (u^2+v^2+4uv)x^2 - 2(u+v)uvx]$$

,

And it is a quartic potential function.
We hope that such a supplement in the manuscript can make this part easier to
understand.
2. Fig. 6: in my opinion the authors should revise this figure, also concering
comments raised by previous reviewers. There is still a discrepancy between the
different form of k, it is not simply readable, and it seems to me that there is
something wrong in plotting the green dots. Please revise.
[REPLY]: Thanks a lot about this. The letters are so similar that it is difficult to
distinguish them. We apologize for this and modify the mistakes in the manuscript
now.
3. Fig. 7: as for Fig. 6 in my opinion the authors should revise this figure. Where is
the gray region in the upper-right corner stated in the caption? Moreover, the authors
should also insert
[REPLY]: This is a mistake. The caption has been corrected as:
*"Figure 7. Statistical results of instability parameters for different sub-sequences*
*lengths. The X-axis is the value of parameter k, and the Y-axis is the statistical*
*frequency for the sub-sequence length of 10a, 20a 30a 40a 50a and 60a."*

4. I'm not sure I completely understood the main purposes of the manuscript as written in the present form. Indeed, it seems to me that the authors state that "By using a piece-wise function, the transition process is stated approximately" (lines 14-15 in the Abstract) and that "Thus, we had proposed a method (Yan et al, 2015, 2016) to study the transition process by using a continuous function" (lines 17-18 in the Abstract) but they are using a piecewise prediction method which is based on a linearization of the logistic equation. So, what would be the main benefit? Why not to directly use the logistic fit for investigating the transition?

[REPLY]: This is a quality comment. We also hope that we can obtain the parameters of the logistic model by fitting the climate time sequence with the logistic model directly. Unfortunately, this is not easy.

The transition process also can be obtained by fitting the climate time sequence with the ramp function which is also called piece-wise function. However, there is no dynamics of the piece-wise function, which means that there is not quantitative relationship during the transition process. The logistic model can provide enough parameters to describe the dynamic process. Next, we analyze how to obtain the parameters of the logistic model.

For a climate time sequence, we don't know when the abrupt change happens, which means that any length of the entire sequence must be taken out to study. That's what we did.

We use the solution, $x = \dfrac{u-v}{1-e^{k(u-v)(t-t_0)}} + u$, of logistic model to fit the segment of the entire time sequence as shown in figure 2.1d. We have the values of variables $x$ and $t$, but it is not easy to know the values of parameters $u, v$ and $k$. Assuming $t_0=0$, we can obtain the value on the right side of the above equation by given values of parameters $u, v$ and $k$. And the Root Mean Square Error Function with $x$ is established,

$$\begin{cases} x'_t = \dfrac{u_m - v_n}{1 - e^{k_o(u_m - v_n)t}} + u_m \\ \xi_{m,n,o} = \left( \dfrac{1}{T} \sum_{t=1,T} (x_t - x'_t)^2 \right)^{\frac{1}{2}} \end{cases},$$

where the parameter $T$ represents the total time of the segment. The parameter $u_m$ represents for any value in the range of the parameter $u$. The parameters $v_n$ and $k_o$ represent for any values in the range of the parameters $v$ and $k$ respectively.

In order to simplify the calculation, the range of parameters needs to be determined in advance. It is obvious that the values of parameters $u$ and $v$ should be between the maximum and minimum values of the sequence. However, the range of the parameter $k$ is unclear. Even if we give the $k$ value a large enough range, it is still difficult for us to obtain enough precision for the parameters. Because if we want to improve the accuracy of the parameters by 10 times, the amount of calculation will be increased by 1000 times.

In figure 2.2, we had proposed a method to divide the segment into three parts. They are two equilibrium states and one transition process. We only need to change the length of each part and perform linear fitting to obtain parameters' values with more
precision by using the following equation,

$$
\begin{cases}
v = \sum_{i=1}^{n_1} x_i / n_1 \\
u = \sum_{i=n_1+n_2+1}^{n} x_i / n_3 \\
h = \sum_{i=n_1+1}^{n_1+n_2} \bar{i} \cdot \bar{x}_i / \sum_{i=n_1+1}^{n_1+n_2} \bar{i}^2
\end{cases}
.
$$

[Figure]

Figure 2.1 Schematic diagram of direct fitting method

[Figure]

Figure 2.2 Schematic diagram of the method we had proposed

Moreover, the authors use the term "tipping point" but they are not
considering/discussing implications of tipping points as well as comparing their
results with established methods for tipping point evaluation (variance, autocorrelation, ..., see papers by Ditlevsen, Lenton, and colleagues). I urge the authors reorganize both the introduction and the conclusion sections to take into account these aspects.

[REPLY]: Thanks for the suggestion about the references. We have added some researches on cusp catastrophe and its early warning in the text.

Minor comments:

- Page 2, line 6: the term "attractors" is too wide in this context, probably it is better to use "fixed points" or "equilibrium states".

[REPLY]: Thanks for the suggestion, we replace "attractors" to be "equilibrium states".

- Page 4, line 9: the term "discontinuous" seems not to be appropriate, should be better "discrete".

[REPLY]: Thanks for the suggestion, we correct the mistake.

- Page 4, line 19: "k as 0.4" should be "k as 0.5". Indeed, in Fig. 1 being u and v fixed then k values are chosen as 0.3 and 0.5, when k=-0.4 is chosen different values for u and v are considered.

[REPLY]: Thanks for noticing these small differences. And they are mistakes. After checked the original data, we find that the $k$ values are 0.4 (the dash gray line) and 0.3 (the black line)for $v$=-1, $u$=2. The $k$ value is -0.4 for $v$=1.0, $u$=2.0. We correct this part as:

*" As shown in figure 2a, parameters u and v being fixed, and setting k as 0.4 ( the dash gray line), the system increasing to the new state costs a shorter time than that setting k as 0.3 (the black line). It is noted that if k<0 ( as v=1.0, u=2.0 and k=-0.4 ), the system decreases from state 2.0 to state 1.0 as the gray line. This states that if the absolute value of k is relatively large, the shorter the transition time of the system, that is, the more unstable(Yan et al, 2016). "*

- Fig. 5: are the authors showing $x_t'$ and not $x_t$? Please confirm.

[REPLY]: We add the caption of figure 5 as:

[revised manuscript text omitted]

---

## Author Response (AR4)

Dear editor and reviewer,

Thanks a lot for the suggestion about the figures.

The Reviewer's comments (text in bold in this document) and our answer (as normal text) are as follows. Parts taken from the manuscript are written in italic font.

**The authors addressed my previous concerns and I think now the**

**manuscript can be accepted for publication. However, I would like to**

**recommend to revise the quality of figures that are not all appropriated. As an**

**example Fig. 6 is not easy to understand and to be readable.**

We have redrawn figure 6 in manuscript as follows.

[Figure]

*Figure 6. Identification of the PDO time sequence and instability parameter k with different sub-sequence lengths.*

*(a) The histogram represents the PDO time sequence (left Y-axis), and the green dots indicate the value of*

*parameter k when the sub-sequence is 20 years (right Y-axis), where the X-axis represents time in year; (b) the*

*start moments of transition processes with different sub-sequence lengths (the red color scatters represent*

*increasing processes, and blue color scatters represent decreasing changes, with deeper colors representing higher*

*values). The green line represents that the value of sub-sequence is 20 years. The X-axis represents the start*

*moment of abrupt change in year, and the Y-axis represents the sub-sequence length in years.*

The original figure 6 was :

[Figure]

.

The X-axis and the Y-axis in figure 6b was exchanged and the scatters with values during (-1, 1) were set to be invisible. A rectangle wireframe was added outside the legend to distinguish it from the scatters. The description in text about figure 6b has been rewritten in line 21 and line 27 page 10.

We hope that figure 6 is clearer and more readable.

Another two corrections are as follows:

1. More details about the author's work unit are added in line 8 page1.

2. One of the foundation ( 2018YFA0606301 ) had been finished, and it is removed now in section Acknowledgements.

[revised manuscript text omitted]